# Loss of the Arp2/3 complex component ARPC1B causes platelet abnormalities and predisposes to inflammatory disease

Walter H.A. Kahr[1,2,3], Fred G. Pluthero[1], Abdul Elkadri[1,4,5], Neil Warner[1,4], Marko Drobac[1,3], Chang Hua Chen[1,3], Richard W. Lo[1,3], Ling Li[1], Ren Li[1], Qi Li[1,4], Cornelia Thoeni[1,4], Jie Pan[1,4], Gabriella Leung[1,4], Irene Lara-Corrales[6], Ryan Murchie[1,4], Ernest Cutz[7], Ronald M. Laxer[8,9], Julia Upton[10], Chaim M. Roifman[10], Rae S.M. Yeung[1,5,8,11], John H. Brumell[1,4,5,12] & Aleixo M. Muise[1,3,4,5]

Human actin-related protein 2/3 complex (Arp2/3), required for actin filament branching, has two ARPC1 component isoforms, with ARPC1B prominently expressed in blood cells. Here we show in a child with microthrombocytopenia, eosinophilia and inflammatory disease, a homozygous frameshift mutation in *ARPC1B* (p.Val91Trpfs*30). Platelet lysates reveal no ARPC1B protein and greatly reduced Arp2/3 complex. Missense *ARPC1B* mutations are identified in an unrelated patient with similar symptoms and ARPC1B deficiency. ARPC1B-deficient platelets are microthrombocytes similar to those seen in Wiskott–Aldrich syndrome that show aberrant spreading consistent with loss of Arp2/3 function. Knockout of *ARPC1B* in megakaryocytic cells results in decreased proplatelet formation, and as observed in platelets from patients, increased ARPC1A expression. Thus loss of ARPC1B produces a unique set of platelet abnormalities, and is associated with haematopoietic/immune symptoms affecting cell lineages where this isoform predominates. In agreement with recent experimental studies, our findings suggest that ARPC1 isoforms are not functionally interchangeable.

[1] Cell Biology Program, Research Institute, Hospital for Sick Children, Toronto, Ontario, Canada M5G 0A4. [2] Division of Haematology/Oncology, Department of Paediatrics, University of Toronto and The Hospital for Sick Children, Toronto, Ontario, Canada M5G 1X8. [3] Department of Biochemistry, University of Toronto, Toronto, Ontario, Canada M5S 1A8. [4] SickKids Inflammatory Bowel Disease Center and Division of Gastroenterology, Hepatology, and Nutrition, Department of Paediatrics, University of Toronto, Hospital for Sick Children, Toronto, Ontario, Canada M5G 1X8. [5] Institute of Medical Science, University of Toronto, Toronto, Ontario, Canada M5S 1A8. [6] Section of Dermatology, Division of Paediatric Medicine, Department of Paediatrics, University of Toronto and The Hospital for Sick Children, Toronto, Ontario, Canada M5G 1X8. [7] Division of Pathology, The Hospital for Sick Children, Toronto, Ontario, Canada M5G 1X8. [8] Division of Rheumatology, Department of Paediatrics, University of Toronto, The Hospital for Sick Children, Toronto, Ontario, Canada M5G 1X8. [9] Department of Medicine, University of Toronto, Toronto, Ontario, Canada M5S 1A8. [10] Division of Immunology, Department of Paediatrics, University of Toronto, The Hospital for Sick Children, Toronto, Ontario, Canada M5G 1X8. [11] Department of Immunology, University of Toronto, Toronto, Ontario, Canada M5S 1A8. [12] Department of Molecular Genetics, University of Toronto, Toronto, Ontario, Canada M5S 1A8. Correspondence and requests for materials should be addressed to W.H.A.K. (email: walter.kahr@sickkids.ca) or to A.M.M. (email: aleixo.muise@utoronto.ca).

In eukaryotic cells, the monomeric ATP-binding protein globular actin (G-actin) is assembled into dynamic filamentous actin (F-actin) in cytoskeletal structures. This process involves a variety of actin-binding proteins[1,2] that sequester/ deliver actin monomers and facilitate the nucleation, elongation, capping, severing, depolymerization and crosslinking of F-actin[3]. Disruption of these processes can result in dysregulation of the actin cytoskeleton, which is associated with metastatic cancer, autoimmune disorders and congenital defects[4]. In Wiskott–Aldrich syndrome (WAS) and X-linked thrombocytopenia (XLT), mutations in WAS (encoding WASP) cause microthrombocytopenia (that is, reduced numbers and size of blood platelets), immunodeficiency, eczema, increased malignancies and autoimmune symptoms including vasculitis and inflammatory bowel disease[5–8]. WASP is one of several nucleation promoting factors that can promote branching of F-actin via the actin-related protein 2/3 complex (Arp2/3), which plays a central role in cell migration, endocytosis, vesicular trafficking and cytokinesis[3,4,9]. Cytoplasmic WASP has an auto-inhibited conformation that is activated due to phosphorylation of tyrosine 291 (refs 10,11) by the Rho family GTPases CDC42 (cell division cycle 42) and NCK1 (non-catalytic region of tyrosine kinase 1). Activated WASP binds Arp2/3 with 2:1 stoichiometry by interacting with the ARP2, ARP3 and ARPC1 subunits, inducing conformational changes that facilitate binding of actin monomers and daughter filament growth[12,13].

The mammalian Arp2/3 complex contains five unique components: ARP2, ARP3, ARPC2, ARPC3 and ARPC4, together with one molecule each of the isoform pairs ARPC1A and ARPC1B, and ARPC5A and ARPC5B. ARPC1A and ARPC1B are located in tandem on human Ch.7 (ref. 14). The isoforms they encode have 68% amino acid sequence identity[15] and both have six WD40 domain repeats predicted to form a β-propeller-fold[3,16]. ARPC1A has been implicated as a regulator of cell migration and invasion in pancreatic cancer[14], while increased ARPC1B has been linked to oral squamous cell carcinoma[17]. Independent of its function in Arp2/3, ARPC1B has also been identified as a centrosomal protein involved in mitosis[18,19]. Total loss of Arp2/3 function is embryonic lethal[20], while its inhibition within cells blocks lamellipodia formation and migration[21,22]. To our knowledge no inherited human deficiency of ARPC1B or other Arp2/3 components has been previously reported.

Here we describe three individuals from two families with homozygous mutations in ARPC1B. Patient 1 with p.Val91Trpfs*30 ARPC1B mutation that results in complete loss of ARPC1B protein and decreased Arp2/3 complex in platelets, leading to microthrombocytopenia, decreased platelet dense granules, defective platelet spreading with prominent filopodia but limited lamellipodia. The patient also has had repeated invasive infections, inflammatory bowel disease, leukocytoclastic vasculitis and eosinophilia. Patients 2 and 3 with a p.Ala105Val ARPC1B mutation that results in minimal ARPC1B protein in their platelets that are microthrombocytes with dense granule deficiencies and similar spreading abnormalities. Patient 2 has leukocytoclastic vasculitis and Patient 3 had intermittent eczematous-like rashes and both have eosinophilia. Thus ARPC1B deficiency in humans results in defective Arp2/3 actin filament branching that is associated with multisystem disease including platelet abnormalities, cutaneous vasculitis, eosinophilia and predisposition to inflammatory diseases.

## Results

### Identification of ARPC1B-deficient patients. We investigated two independent families where patients presented early in life

with failure to thrive, platelet abnormalities, eosinophilia, small vessel vasculitis, eczema and other indicators of inflammatory/ immune disease (Fig. 1a–d, Table 1 and Supplementary Note 1, Clinical information). Whole exome sequencing (WES) of Patient 1 and his parents revealed homozygosity for a novel ARPC1B variant c.269_270dupCT (Supplementary Table 1, Supplementary Data, Fig. 1e,f). This 2 base pair duplication produces a frame shift in exon 4, resulting in a premature stop codon predicted to truncate ARPC1B at amino acid 119. This would yield a product lacking five of the six WD40 repeat domains (Fig. 1g) that form a β-propeller required for interaction of ARPC1B with mother actin filaments[16]. WES of Patient 2 and his parents (Supplementary Table 1, Supplementary Data, Fig. 1f,g) identified homozygosity for two missense ARPC1B variants (c.314C>T and c.712G>A encoding p.Ala105Val and p.Ala238Thr) in this patient and a sibling (Patient 3). Both variants affect WD40 domains; c.314C>T is novel and predicted to be deleterious while c.712G>A is predicted to be neutral (Supplementary Table 1).

Patient 1's parents are consanguineous; thus we focused our initial genetic analysis on Mendelian autosomal recessive mutations. As shown in Supplementary Data, there were no homozygous mutations in any known genes associated with primary immune deficiency, including WASP and WASP interacting protein (WIP), or platelet disorders that could explain the complex phenotype observed in Patient 1. Furthermore, we identified no overlapping compound heterozygous, X-linked, or de novo mutations shared between Patients 1 and 2 (Supplementary Fig. 1). We then focused on novel genes and examined known biological function, known diseases associated with genes, gene expression profiles and available animal models of the candidates described in the Methods section and Supplementary Fig. 1. The critical role of ARPC1B in Arp2/3 function made ARPC1B a viable candidate for the WAS-like disease phenotype observed in both patients, and ARPC1B was the only mutated gene they had in common (see Methods, Identification of ARPC1B mutations). Immunofluorescence (IF) microscopy of skin and intestinal biopsies (Supplementary Fig. 2) indicated loss/reduction of ARPC1B expression in patients. Expression of ARPC1B and other Arp2/3 components in these patients was further studied in a readily accessible cell source provided by blood platelets.

### ARPC1B expression and Arp2/3 complex in platelets. Expression of Arp2/3 components (Fig. 2a) was examined in normal, unaffected family members and patient platelet lysates via immunoblot analysis (Fig. 2b, Supplementary Fig. 3a–c), which showed the absence of ARPC1B in Patient 1, and greatly reduced levels in Patients 2 and 3. Platelet levels of other Arp2/3 components (ARP2, ARP3, ARPC2, ARPC3 and ARPC5; Fig. 2b) and WASP (Fig. 2c) were normal in all patients, and also in unaffected family members (Supplementary Fig. 3c). Recent proteomics data indicate that ARPC1A is present at <6% of the level of ARPC1B in normal platelets[23] (Supplementary Table 2), with isoform levels and ratios differing considerably among cell/ tissue types (Supplementary Table 3). We observed low expression of ARPC1A relative to ARPC1B in normal platelets, and increased ARPC1A in ARPC1B-deficient cells (Fig. 2b,c). Since platelets lack nuclei, this indicates upregulation of ARPC1A in ARPC1B-deficient platelet precursor megakaryocytes (MKs). This parallels the reported upregulation of ARPC1B when siRNA was used to suppress expression of ARPC1A in HeLa cells[24]. The same study also reported, as we saw, that there was no effect of loss of ARPC1B on the expression of other Arp2/3 components.

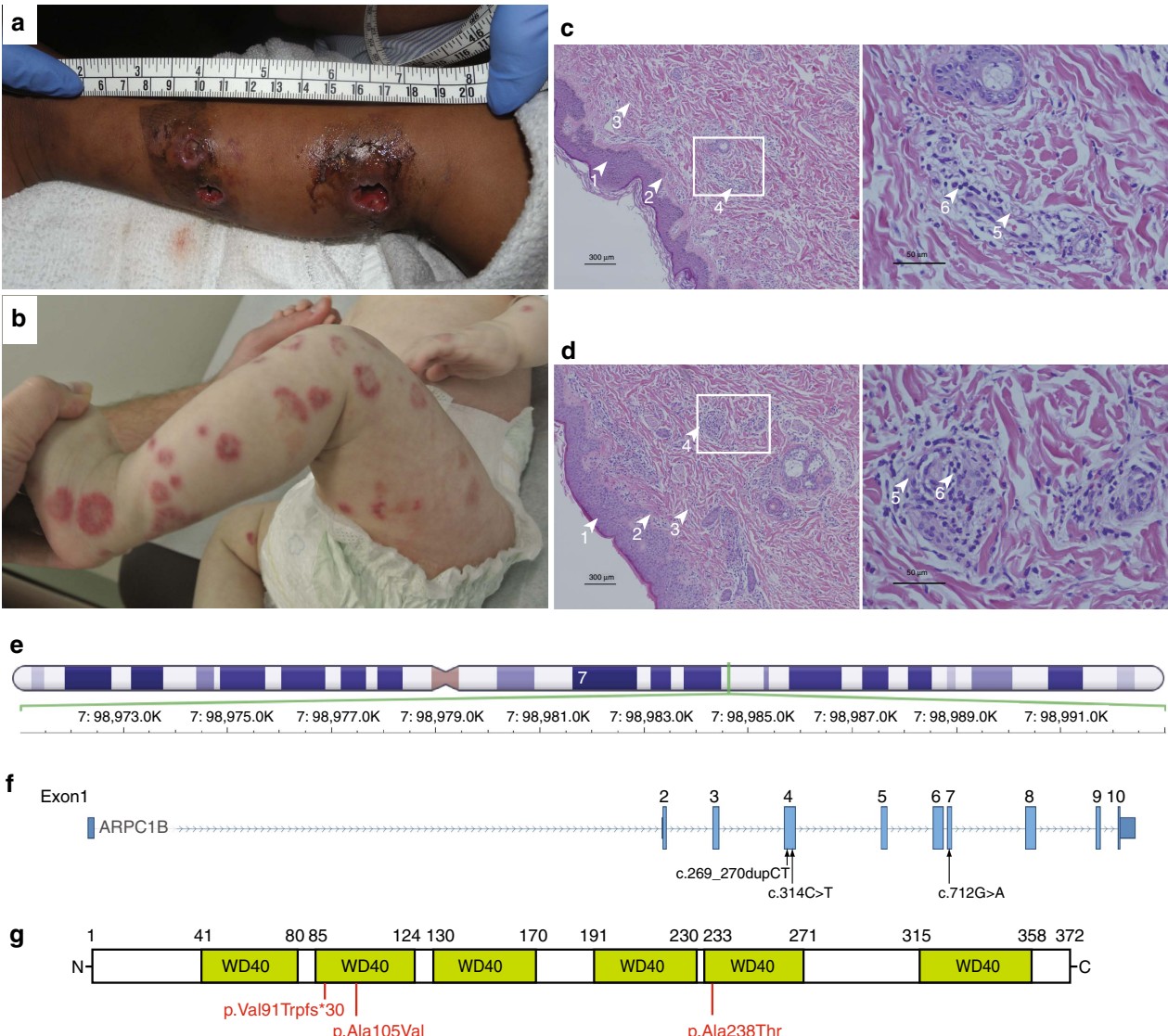

**Figure 1 | Vasculitis pathology and description of *ARPC1B* variants in patients.** Patient 1 (**a,c**) and Patient 2 (**b,d**) developed skin lesions associated with small vessel vasculitis. Patient 1 (**a**) had a complicated ulcerating lesion after a skin biopsy preceded by the more typical skin lesions seen in Patient 2 (**b**). Images **a,b** are not included in the Creative Commons license for the article. Small vessel vasculitis was confirmed by skin biopsies (**c,d**); arrowheads indicate: (1) epidermis, (2) dermal–epidermal junctions, (3) dermis. At low magnification (**c,d**, left panels) areas of leukocytoclastic vasculitis (4) were evident, which at higher magnification (**c,d**, right panels) showed vessel wall destruction (5) and neutrophil infiltration (6). (**e**) *ARPC1B* is located on Chromosome 7 (position numbering relative to GRCh37), immediately preceded by *ARPC1A*. (**f**) Nucleotide positions of identified mutations (black arrows) relative to *ARPC1B* coding exons (accession #: NM_005720.3). Patient 1 is homozygous for c.269_270dupCT, Patient 2 is homozygous for two missense variants (c.314C > T and c.712G > A). (**g**) ARPC1B has 6 WD40 repeat domains forming a β-propeller required for Arp2/3 complex function. The amino acid change caused by the mutation in Patient 1 causes a frame shift predicted to yield a protein lacking the last five WD40 domains; both mutations carried by Patient 2 affect WD40 domains. Adapted from http://smart.embl-heidelberg.de/smart/show_motifs.pl; ARC1B_HUMAN, O15143.

Native gel electrophoresis, which allows multiprotein complexes to be separated under non-denaturing conditions, was used to assess the presence of Arp2/3 in platelet cell lysates. Immunoblotting (IB) detected a band corresponding to Arp2/3 via probing for ARPC1B (Fig. 2d), ARP3 or ARPC5 (Supplementary Fig. 3d). ARP3, ARP2 and ARPC2 were detected with ARPC1B when the Arp2/3 gel band was removed, resolved via denaturing SDS–PAGE, blotted and sequentially probed (Fig. 2d). In contrast, native gel electrophoresis of ARPC1B-null platelet lysate showed barely detectable Arp2/3 (detected via ARPC5 probing; Fig. 2e, left panel). While ARPC1A was not detectable in Arp2/3 from normal platelets, it was detected in the low levels of Arp2/3 present in ARPC1B-null platelets (Fig. 2e, right panel).

**Morphology of ARPC1B-deficient platelets**. Abnormally small platelets (microthrombocytes) in humans have previously only been observed in patients with WAS with complete loss of WASP expression, or in patients with XLT where WASP expression is impaired but not abolished[7,8]. While both conditions are accompanied by thrombocytopenia and bleeding, XLT is associated with minimal or no immunodeficiency[5]. Features of WAS including thrombocytopenia, recurrent infections and eczema were also observed in a patient with loss of WIP, where platelet volume was normal[25]. *WAS* knockout mice have moderate thrombocytopenia and normal-sized platelets[26] while microthrombocytes are observed in mice with MK-specific knockout of the actin-binding protein profilin1 (ref. 27).

**Table 1 | Laboratory parameters of patients.**

| Parameter | Patient 1 | Patient 2 | Patient 3 |
|---|---|---|---|
| Haemoglobin (g l$^{-1}$) | **103** (115–180) | **103** (110–140) | 122 (120–160) |
| White blood cells ($\times 10^9$ l$^{-1}$) | **29** (5.0–20.0) | **23.5** (5.0–12.0) | **13.8** (4.0–10.0) |
| Platelets ($\times 10^9$ l$^{-1}$) | **18** (150–400) | 345 (150–400) | 335 (150–400) |
| Lymphocytes ($\times 10^9$ l$^{-1}$) | 11.9 (2.0–17.0) | **17.4** (4.0–10.5) | 4.18 (1.5–7.0) |
| Eosinophils ($\times 10^9$ l$^{-1}$) | **3.5** (0.7–1.0) | **2.82** (0.05–0.70) | **3.14** (0.02–0.05) |
| ESR (mm h$^{-1}$) | **102** (1–10) | **55** (1–10) | **35** (1–10) |
| *Markers* | | | |
| CD3 (cells µl$^{-1}$) | 1,409 (900–4,500) | 5,609 (2,400–6,900) | 1,484 (700–4,200) |
| CD4 (cells µl$^{-1}$) | 914 (500–2,400) | 4,487 (1,400–5,100) | 992 (300–2,000) |
| CD8 (cells µl$^{-1}$) | 122 (300–1,600) | 1,596 (600–2,200) | 431 (300–1,800) |
| CD19 (cells µl$^{-1}$) | 5,910 (200–2,100) | 8,469 (700–2,500) | 1,545 (200–1,600) |
| CD56 (cells µl$^{-1}$) | 251 (100–1,000) | 932 (100–1,000) | 987 (90–900) |
| *Mitogenic responses (stimulation index percent of control)* | | | |
| PHA | Normal ($>$50) | Normal ($>$50) | Normal ($>$50) |
| αCD3 | Normal ($>$50) | Normal ($>$50) | Normal ($>$50) |
| *Immunoglobulins* | | | |
| IgG (g l$^{-1}$) | 11.6 (4.5–14.3) | 9.7 (1.1–7.0) | 11.9 (5.4–13.6) |
| IgA (g l$^{-1}$) | **6.1** (0.2–1.0) | **5.1** (0.0–0.3) | **4.4** (0.5–2.2) |
| IgM (g l$^{-1}$) | 0.4 (0.2–1.8) | 1.0 (0.2–0.9) | 0.8 (0.4–1.5) |
| IgE (IU ml$^{-1}$) | **1,366** ($<$163) | **414** ($<$25) | **1,799** ($<$90) |
| *Specific antibodies* | | | |
| Anti-tetanus (IU ml$^{-1}$) | 1.73 ($>$0.1) | $>$7.00 ($>$0.1) | 1.21 ($>$0.1) |
| Anti-pneumococcus (mg l$^{-1}$) | $>$270 | ND | ND |
| Isohemagglutinin | αA: 1:64 ($>$1:8) | ND | αA: 1:16 ($>$1:8) |
| | αB: 1:64 ($>$1:8) | | αB: 1:8 ($>$1:8) |
| *Auto-antibodies* | | | |
| ANA | **1:160** (Neg) | **1:160** (Neg) | Neg (Neg) |
| ANCA | **Pos** (Neg) | **Pos** (Neg) | **Pos** (Neg) |
| TRECS (copy no/3 µl) | $>$**2,000** ($>$400) | **1,239.5** ($>$400) | **1,636** ($>$400) |

Normal values per age group are shown in brackets. Results outside of normal range in bold.

Thin section transmission electron microscopy (TEM) was used to examine platelets from the ARPC1B-null patient, a WASP-null patient and a normal donor. All had platelets containing typical cellular structures including mitochondria and α-granules (Figs 3a and 4). ARPC1B-null and WASP-null platelets showed a propensity towards small size (Fig. 4), and examination via IF microscopy (Fig. 3b) confirmed these platelets to be small and dysmorphic compared to normal. A comparison of circumferential tubulin ring diameters (Fig. 3c) confirmed that both ARPC1B-null and WASP-null platelets are significantly smaller than normal, and do not differ significantly from each other. ARPC1B-null platelets can thus be classified as micro-thrombocytes, as can ARPC1B-deficient platelets from Patients 2 and 3 (Fig. 5a). A significant proportion ($\sim$20%) of ARPC1B-null platelets shared dysmorphic features with WASP-null platelets (Fig. 3d,e) that included odd shapes, collapse/loss of circumferential microtubule coils and highly variable P-selectin and thrombospondin-1 content (both indicators of α-granules). As has been reported for WAS platelets[28], whole-mount TEM[29] revealed a reduction/absence of calcium-rich platelet dense granules in ARPC1B-null and -deficient platelets (Fig. 5b,c). Clinical lumi-aggregometry analysis of platelets from Patients 1 and 2 confirmed decreased dense granule ATP release (0.16 and 0.19 nmol respectively; normal range 0.29–1.93 nmol). Platelet aggregation investigations with collagen, SFLLRN, arachidonic acid, ristocetin and ADP were normal for both patients.

**Spreading behaviour of ARPC1B-deficient platelets**. Actin rearrangements within cells can produce several types of membrane protrusions[30]. Spindle-like filopodia are driven outwards by parallel actin filaments generated by formin protein family members[30,31], while broad lamellipodia involve branching of actin networks and elongation of filaments[32]. Platelets spreading on a surface typically extend filopodia before producing lamellipodia[33,34]. This process proceeds normally in WASP-deficient platelets, because the nucleation promoting factor required for activating Arp2/3 (refs 35,36) during lamellipodia formation is WAVE/SCAR[37,38] rather than WASP.

We examined the consequences of ARPC1B deficiency for platelet spreading on fibrinogen-treated surfaces using high-resolution fluorescence microscopy and scanning electron microscopy to monitor cell morphology and intracellular localization/distribution of tubulin, F-actin and Arp2/3 components. Comparisons of both washed platelets (Fig. 6) and platelets in plasma (Fig. 7) showed that maximally spread cells from normal donors formed typical near-circular lamellipodia. As expected from experimental observations[36], normal platelet lamellipodia had peripheral localization of F-actin and ARPC5 (Fig. 7), and also displayed prominent F-actin stress fibres and podosome-like nodules. In contrast, maximally spread ARPC1B-null and ARPC1B-deficient platelets typically formed spiky structures with tubulin-rich tips (Figs 6 and 7) that contained fewer and often elongated F-actin fibres and showed little evidence of podosome-like nodule formation. The spread

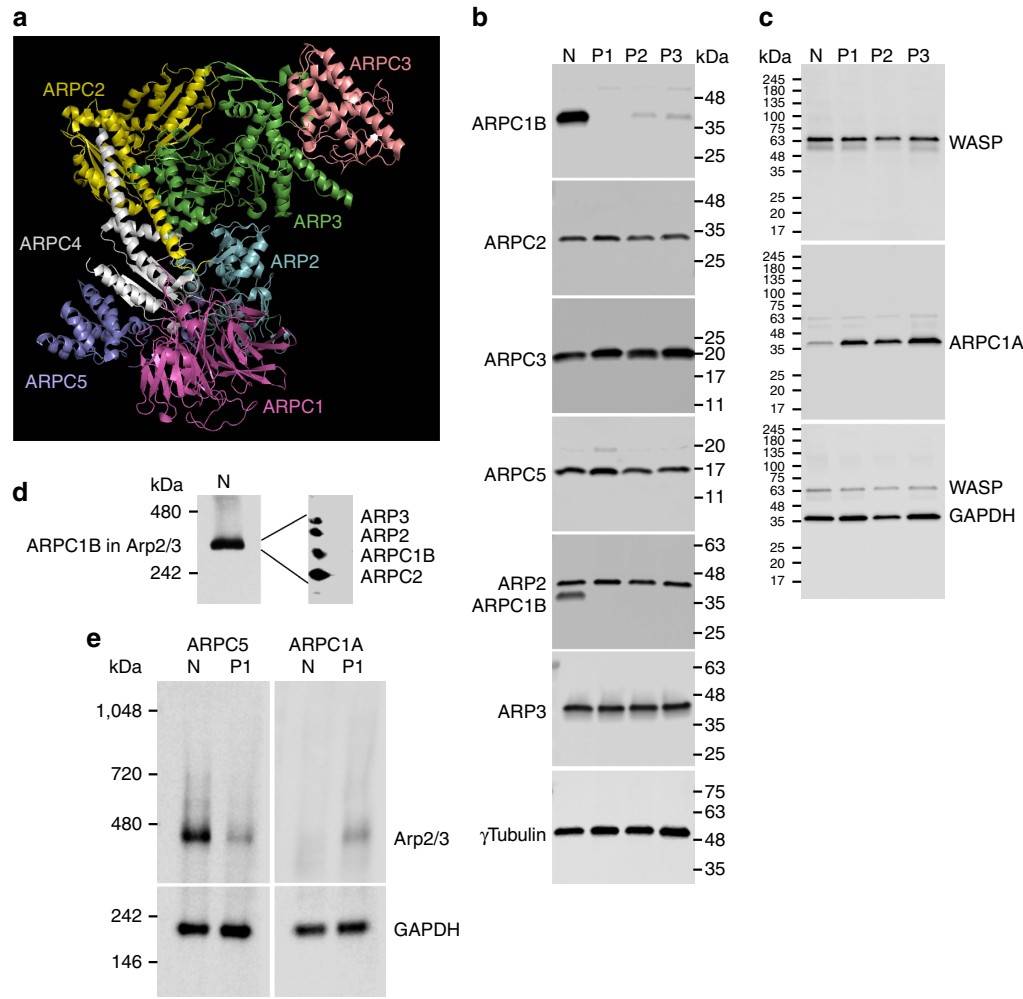

**Figure 2 | ARPC1B and Arp2/3 complex are deficient in *ARPC1B* mutant platelets.** (**a**) Crystal structure of mammalian Arp2/3 showing relative location of component proteins from the protein data bank (PDB1K8K, http://www.rcsb.org/pdb/explore/explore.do?structureId=1K8K; Robinson *et al.*[16]). (**b**) Immunoblot analysis of platelet lysates showed ARPC1B to be absent in Patient 1 (P1) and greatly reduced in Patients 2 and 3 (P2, P3) compared to normal (N), while levels of ARPC2, ARPC3, ARPC5, ARP2 and ARP3 were normal (gamma tubulin used as loading control, see Supplementary Fig. 3a). (**c**) Platelet ARPC1A was increased in all three patients relative to normal, while WASP expression was equivalent (GAPDH used as loading control). (**d**) Immunoblot analysis of a normal platelet lysate after native gel electrophoresis showed a band corresponding to Arp2/3 complex detected by probing for ARPC1B (shown) or other Arp2/3 components (Supplementary Fig. 3d). This band was resolved on a second dimension SDS–PAGE gel, and immunoblotting confirmed the presence of Arp2/3 components (ARPC1B, ARPC2, ARP2 and ARP3 shown). (**e**) Immunoblotting of platelet lysates after native gel electrophoresis for ARPC5 (left) showed a greatly reduced level of Arp2/3 in Patient 1 (P1) platelets relative to normal (N). ARPC1A (right) was detected in the Arp2/3 complex in Patient 1 (P1) but not in normal (N) platelet lysate (native GAPDH tetramer used as loading control).

platelet surface area was significantly reduced in ARPC1B-null and -deficient platelets compared to normal (Supplementary Fig. 4). Allowing ARPC1B-null and -deficient platelets more time to spread did not alter their observed behaviour. These observations indicate a profound loss of actin branching required for lamellipodia formation[30,32] in ARPC1B-deficient platelets, despite their increased ARPC1A content. This is consistent with experimental observations that isoforms of Arp2/3 containing ARPC1B are significantly better than complexes containing ARPC1A at promoting the rapid assembly of stable branched actin networks[24].

**Proplatelet formation in ARPC1B knockout megakaryocytic cells.** The formation of proplatelets by MKs is a key stage in platelet development, which can only proceed to completion in the presence of shear flow[39,40]. The actin cytoskeleton has

been linked to the adhesion of proplatelet-forming MKs to the extracellular matrix[41] and in the bifurcation of proplatelets, which increases the number of tips that give rise to platelets[42]. Reduced platelet size and thrombocytopenia are the most common findings in WAS and XLT patients[5]. The cellular mechanisms of WAS-associated microthrombocytopenia are not fully understood. Bone marrow MK numbers are normal in most WAS patients[5,43], but proplatelet formation may occur prematurely, hampering platelet release into the bloodstream[44,45]. Peripheral destruction of platelets in the spleen may also be involved, since splenectomy can correct platelet count and size[46,47].

We observed adequate MK numbers and normal morphology in a bone marrow biopsy sample from Patient 1 (Fig. 8). This indicates that MK depletion is not a cause of thrombocytopenia, although the possibility of abnormal proplatelet formation cannot be ruled out. Platelet counts were always low in Patient 1, low

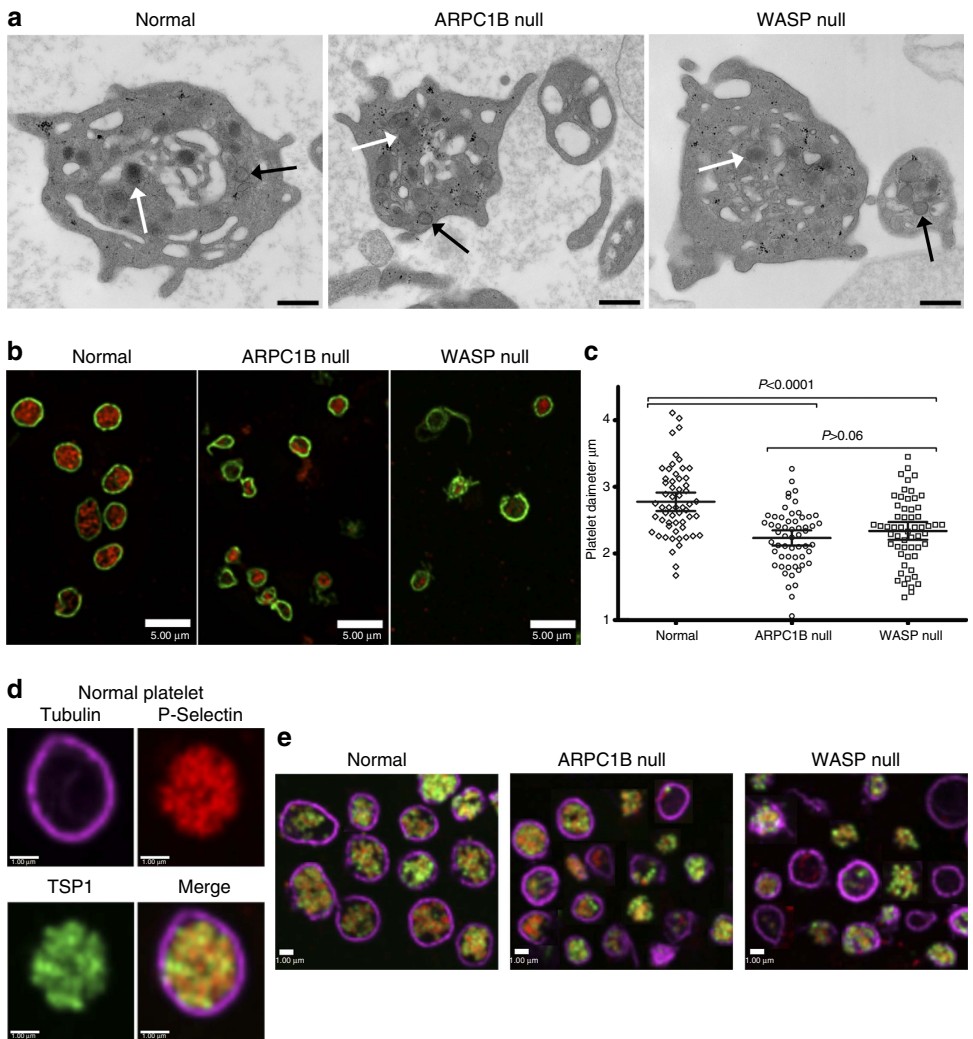

**Figure 3 | ARPC1B-null and Wiskott–Aldrich syndrome platelets are similar.** (**a**) Transmission electron microscopy (TEM) imaging of sections from normal donor, ARPC1B-null (Patient 1) and WASP-null patient platelets show typical structures including α-granules (white arrows), mitochondria (black arrows) and internal membrane systems (bars = 500 nm; see Fig. 4 for further comparisons). (**b**) Immunofluorescence microscopy of cells stained for circumferential ring tubulin (green) and α-granule membrane P-selectin (red) shows that ARPC1B-null and WASP-null platelets appear small and dysmorphic compared to normal (bars = 5 μm). (**c**) A comparison of circumferential ring diameter distributions confirmed that both ARPC1B-null and WASP-null platelets are significantly smaller than normal ($P < 0.0001$, unpaired t-test; $n = 57$ cells; 95% confidence intervals shown) and do not differ significantly from each other. Equivalent comparisons of all patients with WASP-null and normal control platelets is shown in Fig. 5a. (**d**) The morphology of a normal platelet showing the typical subcellular locations of tubulin (magenta), P-selectin (red) and α-granule cargo thrombospondin-1 (TSP1, green; bars = 1 μm). (**e**) Collections of representative cells illustrate the range of abnormal features observed in ARPC1B-null and WASP-null platelets, including small size and the variable absence of internal contents (bars = 1 μm).

(for example, $79 \times 10^9 \, l^{-1}$) to normal in Patient 2, and normal in Patient 3 (Table 1). The microthrombocytes we observed in all three patients suggest that decreased ARPC1B in MKs affects the cytoskeletal dynamics of platelet formation[42,48] sufficiently to influence platelet size. The thrombocytopenia observed in Patient 1 indicates that total loss of ARPC1B leads to depression of platelet production and/or increased clearance.

To explore the potential impacts of loss of ARPC1B expression on platelet production by ARPC1B-null MKs, we used imMKCL cells, a stable immortalized MK progenitor. Unlike most mega-karyocytic cells, imMKCL cells can be stimulated by thrombo-poietin to generate proplatelet-producing cells in culture[49]. imMKCL lines lacking functional *ARPC1B* were made using CRISPR/Cas9 gene deletion. Loss of ARPC1B expression in an *ARPC1B* knockout line was confirmed via IB (Fig. 9a), which also detected increased ARPC1A expression relative to wild-type cells,

paralleling our observations in ARPC1B-null platelets (Fig. 2c). We examined the abilities of thrombopoietin-induced wild-type and ARPC1B-null imMKCL cells to form proplatelet-like extensions in culture, and observed that ARPC1B knockout cells formed proplatelets much less frequently than wild-type cells (Fig. 9b). In addition, while some wild-type imMKCL cells were observed to form multiple proplatelets containing tubulin and branched actin filaments (Fig. 9c), this was not seen in ARPC1B-null cells.

It is likely that the decreased ability of ARPC1B-null imMKCL cells to form proplatelets in culture reflects a similar phenotype in ARPC1B-null MKs, resulting in the thrombocytopenia seen in Patient 1. Since the generation of normal platelets cannot be studied in culture, as the bloodstream is required for final maturation[40], it is not possible to draw definitive conclusions from this experiment regarding the presence of small platelets

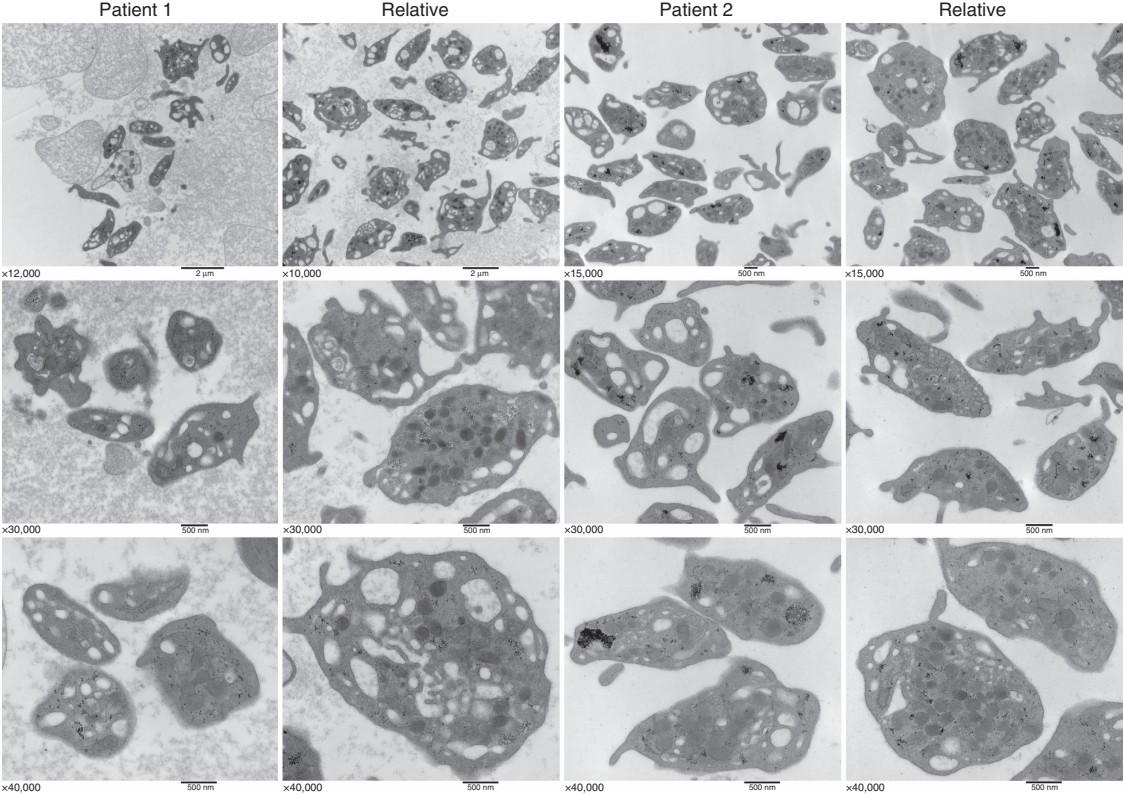

**Figure 4 | Transmission electron microscopy of platelets from ARPC1B-deficient patients and normal relatives.** Electron micrographs of fixed platelet sections taken at magnifications ranging from ×10,000 to ×40,000 (magnification on each panel; bars = 500 nm, top left two panel bars = 2 μm) indicate the presence of generally small and morphologically variable platelets in patient samples (see also Fig. 3). Dense granules were evaluated by whole mount transmission electron microscopy (Fig. 5b,c).

with structural abnormalities observed in our patients (Fig. 3e). It is reasonable to propose that MKs with altered actin dynamics produce small and dysmorphic platelets, and/or that platelets with altered actin dynamics have increased susceptibility to deformation as they circulate. Our results indicate that, as we observed with platelet spreading (Figs 6 and 7), upregulation of ARPC1A in ARPC1B-null imMKCL cells had little or no effect on restoring Arp2/3 functions required for proplatelet formation. This is in keeping with what we would expect to see in proplatelet formation in the absence of ARPC1B.

### Discussion

The Arp2/3 complex is ubiquitous in eukaryotic cells[3] and essential for the survival of multicellular organisms[20]. It has been reported that ARPC1B is both an activator and substrate of Aurora A kinase which is critical in the maintenance of mitotic integrity in mammalian cells[18]. Our results indicate that near-complete loss of ARPC1B expression results in platelet abnormalities including microthrombocytes and spreading defects, and also eczema, leukocytoclastic vasculitis, eosinophilia and elevated IgA and IgE. Elevated IgE levels are also seen in hyper IgE immune deficiency syndrome patients[50], where the pathological effects of elevated IgE are poorly understood but likely involve several immune pathways, including increased Th2 cytokine production[51]. ARPC1B-deficient Patients 2 and 3 have normal platelet numbers, while ARPC1B-null Patient 1 has persistent thrombocytopenia. Our experiments with imMKCL cells indicate that this is likely linked to decreased proplatelet production by ARPC1B-null MKs. It may also be that these patients have increased rates of peripheral platelet clearance

as observed in WAS[5], which may be associated with their elevated IgA levels. If so, this would exacerbate the consequences of low platelet production by ARPC1B-null MKs.

Our results indicate that ARPC1B deficiency is associated with severe multisystem disease including recurrent infections, inflammatory changes in the intestine (crypt distortion with severe eosinophilic infiltration) and elevated autoimmunity markers (anti-nuclear and anti-neutrophil cytoplasmic antibodies; Table 1). This multi-system pathology is similar to that produced by complete loss of WASP expression[5], and it is consistent with the predominant expression of ARPC1B in haematopoietic/immune cells (Supplementary Table 3). These include B- and T-lymphocytes (including T-regulatory cells and natural killer cells), antigen-presenting dendritic cells, monocytes/macrophages and neutrophils. All of these cells require coordinated actin dynamics for development, migration, recruitment, signalling and activation of innate and adaptive immune responses[52].

We observed considerable heterogeneity in the manifestation of disease among the patients we studied. All three had failure to thrive, platelet abnormalities, eosinophilia, eczema and other indicators of inflammatory/immune disease including elevated IgA and IgE, and anti-neutrophil autoantibodies. Only Patient 1 had chronic infections and colitis. Patient 3 had a history of eczema-like rash from birth, although not severe enough to be biopsied, and it is unclear if underlying vasculitis is associated with this rash, as was seen in Patients 1 and 2. In addition to *ARPC1B* mutations, there are likely to be other genetic and/or environmental factors associated with the different disease phenotypes observed among patients. It is nevertheless tempting to propose that as with thrombocytopenia, the milder immune

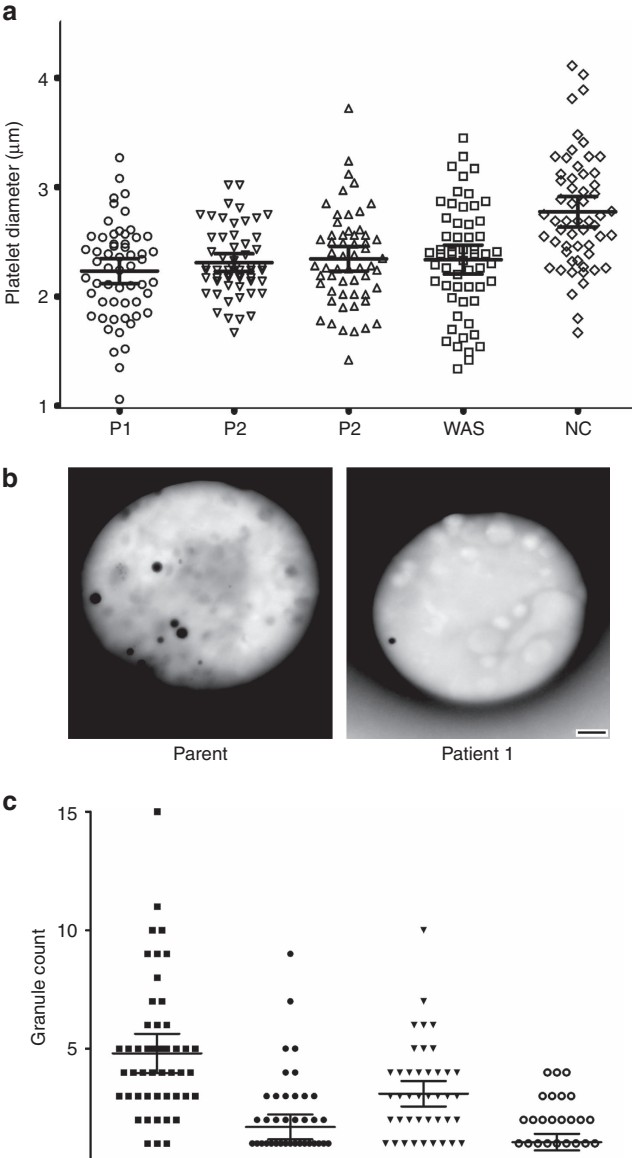

**Figure 5 | ARPC1B-deficient platelets are microthrombocytes containing decreased numbers of dense granules. (a)** A comparison of platelet circumferential ring diameter distributions for all three ARPC1B mutant patients (P1, P2, P3) and a WASP-null patient (WAS) shows that all are significantly smaller (that is, microthrombocytes) compared to platelets from a normal donor (NC; $P < 0.0001$, unpaired $t$-test; $n = 57$ platelets, means and 95% confidence intervals shown by bars). No significant differences were observed among patients. **(b)** TEM of platelet whole mounts from an unaffected parent (left) and Patient 1 (right), showing dense granules detectable as dark spots (bar = 500 nm; × 25,000). **(c)** Dense granule counts derived from whole-mount TEM imaging of platelets from Patient 1 (P1), Patient 2 (P2) and a WASP-null patient (WAS) were significantly lower than for cells from an unaffected relative (F1); Mann–Whitney test $P < 0.0008$, $n = 50$ platelets. Means for Patient 1 and WASP null were not significantly different ($P = 0.07$), while the mean for Patient 2 was significantly higher than for Patient 1 ($P < 0.0001$). Means and 95% confidence intervals shown by bars.

manifestations observed in Patients 2 and 3 may be attributable to their residual ARPC1B expression compared to Patient 1. This would parallel the milder phenotype observed in XLT that is attributed to residual WASP expression[5].

As with WAS[11,52], many questions remain regarding the mechanisms associated with the abnormalities we have observed in patients with absent/reduced ARPC1B expression. Our observations that the loss of ARPC1B has profound consequences for platelet spreading, and most likely for MK proplatelet production, are consistent with cellular phenotypes that would be expected with the loss of Arp2/3 function in these cells. The effects of ARPC1B loss would presumably be most severe in cells/tissues where it is the predominant isoform present in functional Arp2/3, since as we observed in platelets and MKs, compensatory upregulation of ARPC1A has little effect. This is likely due to cell lineage-specific variations in Arp2/3 assembly and/or function[24].

While it is difficult to connect our experimental observations to the entire spectrum of disease observed in ARPC1B-deficient patients, Arp2/3-driven actin polymerization has recently been reported to be essential for several relevant cellular, physiological and developmental processes. These include cell secretion[53], phagocytosis[54,55], autophagy[56], migration[57,58], haptotaxis[59], focal adhesions[60] and intracellular tight junctions required for epidermal barrier formation[61], vesicle trafficking and transcytosis in the small intestine[62]. With regards to immunity and inflammation, Arp2/3 function has been reported to be critical for the formation of immune cell synapses[63,64] and T-regulatory cell function, which is aberrant in WAS patients leading to a high susceptibility to develop Th2-mediated food allergies[65]. Genetic correction of induced pluripotential stem cells from WAS patients demonstrated the restoration of defective natural killer and T-lymphoid cell development and function, confirming the critical role of WASP[66]. Given the central role that Arp2/3 plays in so many processes, it is reasonable to expect that ARPC1B deficiency may be associated with a broad range of developmental and immune defects. Our results also point to the possibility that gene variants affecting other Arp2/3 components may be associated with human disease.

## Methods

**Subjects.** All experiments were carried out with the approval of the Research Ethics Board at the Hospital for Sick Children, Toronto, Canada. Informed consent to participate in research was obtained from all participants. A copy of the consent is available on the interNational Early Onset Paediatric IBD Cohort Study (NEOPICS) website at http://www.neopics.org/study-documents.html. Patients were consented to the registry and tissue bank of the Canadian Centre for Primary Immunodeficiency. Sequencing of the patient with Wiskott–Aldrich syndrome (WASP null) was done by PreventionGenetics (Marshfield, WI, USA) for clinical diagnosis. A hemizygous missense variant c.256C > T in the WAS gene was predicted to result in the amino acid substitution p.Arg86Cys, previously documented to cause WAS[67].

**Genomic sequencing data analysis and validation.** WES was performed at the Centre for Applied Genomics, Hospital for Sick Children, Toronto, Canada. Exome library preparation was performed using the Ion Torrent AmpliSeq RDY Exome Kit following the manufacturer's recommended protocol. In brief, 100 ng of DNA quantified by Qubit DNA HS or BR assay was used in the target amplification under the following conditions: 99 °C for 2 min, followed by ten cycles at 95 °C for 15 s and 60 °C for 16 min, and final hold at 10 °C. Incorporated primer sequences were partially digested using a proprietary method. Ion Torrent Proton adapters were ligated to the amplicons at 22 °C for 30 min followed by 72 °C for 10 min, and the library was purified with Agencourt Ampure XT Beads. Libraries were quantified by qPCR, and 7 pM used for sequencing on an Ion Torrent Proton Sequencer using a PI chip V2 following the manufacturer's protocol. All data were aligned to the hg19/GRCh37 reference genome and quality trimmed via Ion Torrent Suite Version 4.2.

SNP and Variation Suite Version 8.1 (Golden Helix) and VarSeq Version 1.1 (Golden Helix) were used for data analysis. After importing the variant call files of each member of the family trio (patient and parents), variants were organized by pedigree. Rare (minor allele frequency (MAF) < 1%) variants were filtered using the 1,000 genomes Variant Frequencies (Phase 1), the Exome Aggregation Consortium (ExAC) Variant Frequency database Version 0.3 (Cambridge, MA, USA) and the NHLBI Exome Sequencing Project (ESP) V2 Exome Variant Frequencies. Variants were classified according to whether they were deemed to be coding and non-synonymous and unclassified variants

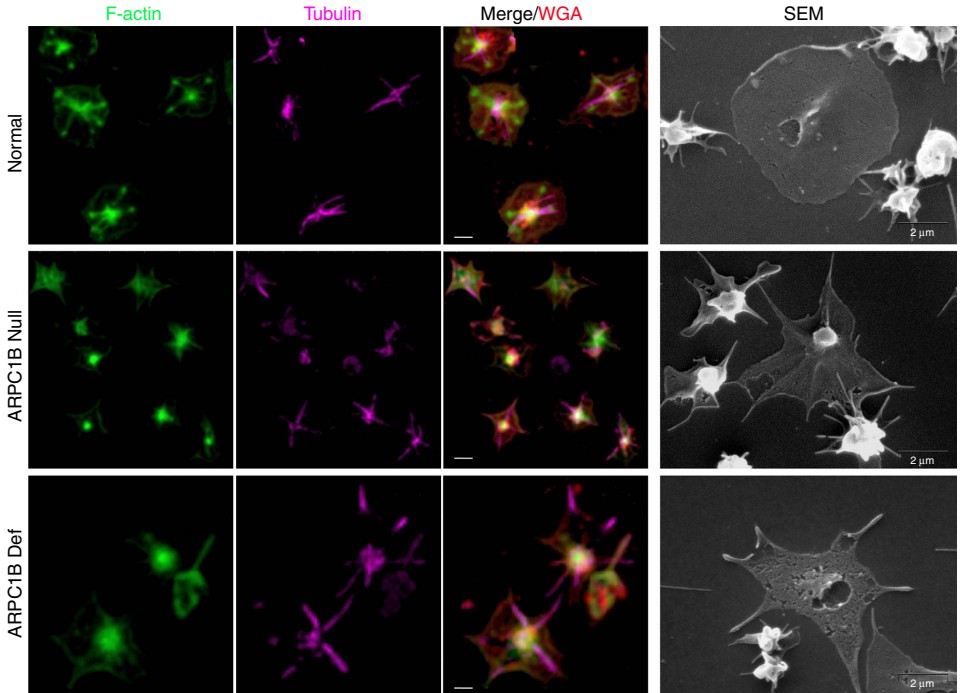

**Figure 6 | Abnormal spreading of ARPC1B-deficient platelets on fibrinogen.** Washed platelets were allowed to spread on fibrinogen-treated coverslips for 45 min prior to fixation and imaging by SEM (right column; bars = 2 μm), or by spinning disc laser fluorescence confocal microscopy (left columns; 3D renders of deconvolved z-series, bars = 1 μm) after surface staining with wheat germ agglutinin (WGA; red), staining for F-actin with phalloidin (green) and immunostaining for alpha tubulin (magenta). Maximally spread platelets from a normal donor (top row) typically show fully formed lamellipodia with thin tubulin filaments and multiple F-actin filaments intersecting with podosome-like actin nodules. In contrast, maximally spread platelets from both ARPC1B-null (middle row) and ARPC1B-deficient (bottom row) patients tend to form spiky filopodial-lamellipodial structures lacking podosomes that sometimes show elongated actin filaments (see also Fig. 7).

were then scored using the database for non-synonymous functional predictions (dbNSFP 2.8), filtering out variants found to have no damaging score (Polyphen2, SIFT, MutationTaster, MutationAssessor, FATHMM). As well, dbNSFP scores variants with conservation scores (PhyloP and GERP + +).

Sanger sequencing was performed to validate mutations identified by WES. The following primers were used to sequence *ARPC1B* Exon4 (forward 5′-GCAGA TACAGCTTCCACC-3′ and reverse 5′-CCCTAACAGCCCACTC-3′) and *ARPC1B* Exon7 (forward 5′-GCTGAGAGTACAGGTGCG-3′ and reverse 5′-CCTGCTGTGACCACACAC-3′).

**Identification of *ARPC1B* mutations.** *Patient 1.* WES of Patient 1 and parents resulted in the identification of 50,020 variants relative to the reference genome Hg19. In all, 31,997 were considered high quality, passing thresholds for genotype quality and read depth. A total of 2,853 were considered rare variants with a minor allelic frequency less than 1% (MAF < 0.01) using three databases: NHLBI ESP6500SI-V2, 1,000 Genomes and ExAC v.0.3. 1,046 variants were predicted to be damaging, as they were classified as missense mutations or predicted to cause loss of function. From this list we identified 14 non-synonymous homozygous variants (Supplementary Data) inherited in an autosomal recessive manner. The *ARPC1B* variant for which Patient 1 is homozygous is a two base pair duplication (c.269_270dupCT; Fig. 1f) causing a frame shift in exon 4 at position 91 and a premature stop codon, with a predicted truncated 119 amino acid protein lacking five of the six WD40 domains (Fig. 1g) required for formation of the functional p40/ARPC1 β-propeller[16]. c.269_270dupCT is a novel variant and the only one predicted to cause loss of function by RefSeq version 105v2 (Supplementary Table 1); it was validated by Sanger sequencing and independent WES.

*Patient 2.* WES of Patient 2 and parents resulted in the identification of 51,866 variants. In all, 35,114 were considered high quality, passing thresholds for genotype quality and read depth. A total of 1,704 were considered rare variants with a minor allelic frequency less than 1% (MAF < 0.01; see above). In all, 676 variants were predicted to be damaging as they were classified as missense mutations or caused loss of function. From this list we identified nine non-synonymous homozygous variants (Supplementary Data) inherited in an autosomal recessive manner. Two homozygous *ARPC1B* missense variants were found near the same region of the mutation identified in Patient 1 (c.314C > T and c.712G > A encoding p.Ala105Val and p.Ala238Thr; Fig. 1f). Analysis using the ExAC, NHLBI, ESP and 1,000 Genomes databases revealed the c.712G > A variant to be rare and predicted to be benign, while c.314C > T is novel and predicted to be damaging (Supplementary Table 1), disrupting the second ARPC1B WD40 domain

and thus the p40/ARPC1 β-propeller. Both variants were validated using Sanger sequencing.

Patient 1's parents are consanguineous, and he has a very severe and complex disease. Therefore we focused our initial genetic analysis on Mendelian autosomal recessive mutations with a homozygous inheritance pattern that could explain his disease. As shown in Supplementary Data no homozygous mutations were detected in predicted pathogenic variants for known genes associated with immune deficiency (including WASP and WIP) or platelet disorders. Also, no overlapping compound heterozygote mutations, X-linked mutations, or *de novo* mutations were shared by Patient 1 and Patient 2. We then focused on novel genes and examined known biological function, known diseases associated with genes, gene expression profiles and available animal models of the candidates outlined in Supplementary Information. The only gene that fit the disease profile observed in Patient 1 was *ARPC1B*, since the role of ARPC1B as the WASP-binding component of the Arp2/3 complex pointed to a WAS-like phenotype. In Patient 2, with a similar spectrum of disease, WES also identified *ARPC1B* as the only viable candidate, and this was the only gene found to be mutated in both Patients 1 and 2 (see Venn diagram Supplementary Fig. 1).

**Antibodies.** Anti-human protein antibodies used for platelet and imMKCL cell IB and IF staining were as follows: rabbit polyclonals to ARPC1A (Sigma-Aldrich, HPA004334; dilutions 1/250 or 1/500 for IB, 1/25 for IF), ARPC1B (Sigma-Aldrich, HPA004832; dilutions 1/100 or 1/500 for IB, 1/100 for IF), ARPC2/p34ARC (Millipore, 07-227; dilution 1/1,000 for IB), ARPC3 (Sigma-Aldrich, HPA006550; dilution 1/1,000 for IB), ARPC5/p16ARC (Abcam, ab118459; dilutions 1/1,000 for IB, 1/100 for IF) and α-tubulin (Cell Signaling, 11H10; dilutions 1/1,000 for IB, 1/100 for IF); rabbit monoclonals to ARP2 (Abcam, ab128934; dilution 1/1,000 for IB), ARP3 (Abcam, ab151729; dilution 1/1,000 for IB) and WASP (Cell Signaling, D9C8; dilution 1/1,000 for IB); mouse monoclonals to γ-tubulin (Thermo Scientific, 4D11), α-tubulin (Sigma-Aldrich, Clone B-5-1-2; dilutions 1/1,000 for IB, 1/100 for IF), CD61 (Dako, Clone Y2/51; dilution 1/200 for IF), TSP1 (R&D Systems, clone 301221; dilution 1/100 for IF) and GAPDH (Millipore, MAB374; dilution 1/5,000 for IB), and a goat polyclonal to P-selectin (Santa Cruz, sc-6941; dilution 1/100 for IF).

**Platelet immunoblot and native gel electrophoresis analysis.** Blood from patients and normal controls was collected by venipuncture with 3.2% sodium citrate anticoagulation and centrifuged (150g, 15 min) before collection of platelet-rich plasma (PRP), from which platelets were pelleted via centrifugation

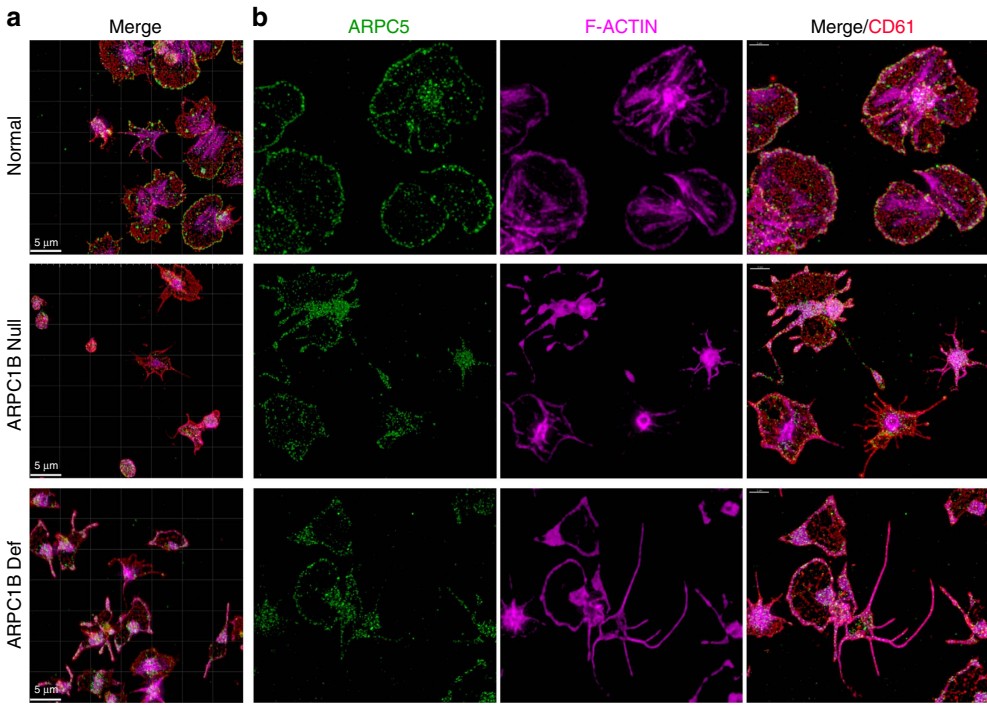

**Figure 7 | Spreading behaviour and distribution of ARPC5 and actin in normal and ARPC1B-deficient platelets.** Platelets in platelet-rich plasma were allowed to spread on fibrinogen for 30 min before fixation, staining and imaging by laser fluorescence structured illumination microscopy. (**a**) Each set of panels shows a wide field image (bars = 5 μm) and (**b**) higher magnification images of representative spread platelets stained for CD61/fibrinogen receptor (red), ARPC5 (green) and F-actin (magenta; bars = 2 μm). Most normal platelets show circular lamellipodia, peripheral ARPC5 and F-actin present at the lamellipodial edge and in stress fibres and nodules (see also Fig. 6). In contrast, at his time point many ARPC1B-null platelets (from Patient 1) have not spread, or show abnormal shapes and subcellular distributions of ARPC5 and F-actin (note: close up panels are collages necessitated by the sparse distribution of spread platelets; all others are cropped fields). ARPC1B-deficient platelets (from Patient 2) also show limited spreading and unusual spread morphologies, including extremely long filopodia containing long unbranched actin fibres.

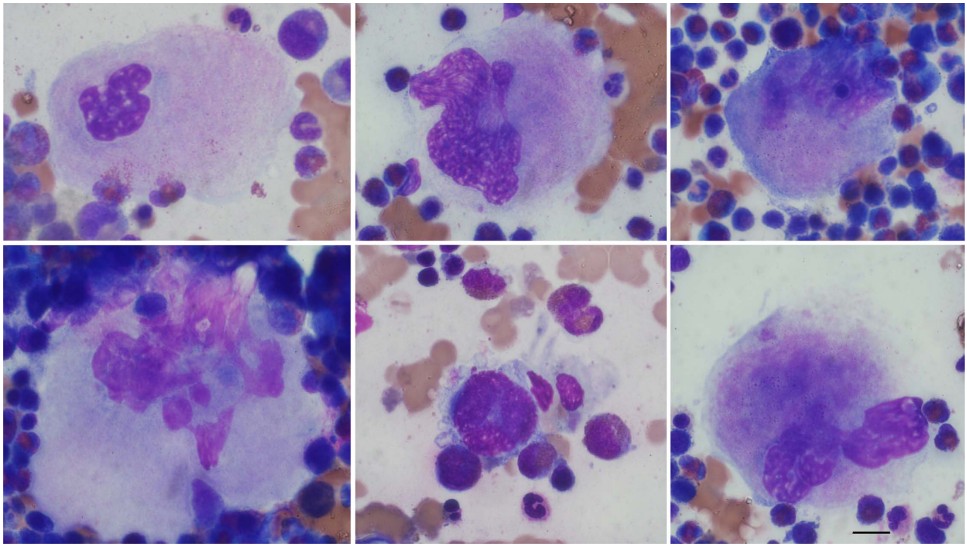

**Figure 8 | Bone marrow megakaryocyte morphology from Patient 1.** Diagnostic bone marrow aspirate showing normal megakaryocytes with multilobed nuclei and large cytoplasm indicating normal morphology. Images were acquired from marrow smear preparations after Wright–Giemsa staining. Magnification × 60, bar = 10 μm.

(1,000g, 10 min). Prior to obtaining lysates, platelets were washed twice by resuspension in phosphate-buffered saline (PBS) buffer adjusted to pH 6.1 with ACD (PBS/ACD) and pelleting. For direct IB platelets were resuspended at 10⁹ per ml in PBS plus 2 × protease inhibitor (Roche Complete EDTA-free, Roche Diagnostics) and lysed with Triton-X100 (0.5%). For native gel electophoresis, washed platelets were resuspended as above with added

phosphatase inhibitor (Roche PhosSTOP) and lysed by sonication via two 5 s pulses 1 min apart in a Heat Systems Sonicator Ultrasonic Processor XL XL2010 (Farmingdale, NY, USA) set at 20 V and amplitude 2. All lysates were cleared by centrifugation at 21,000g for 2 min after which the supernatants were retained and analysed for protein content by IB[68] or for Arp2/3 complex via blue Native gel electrophoresis, where sonicated platelet lysates were applied to a 4–16% Bis-Tris

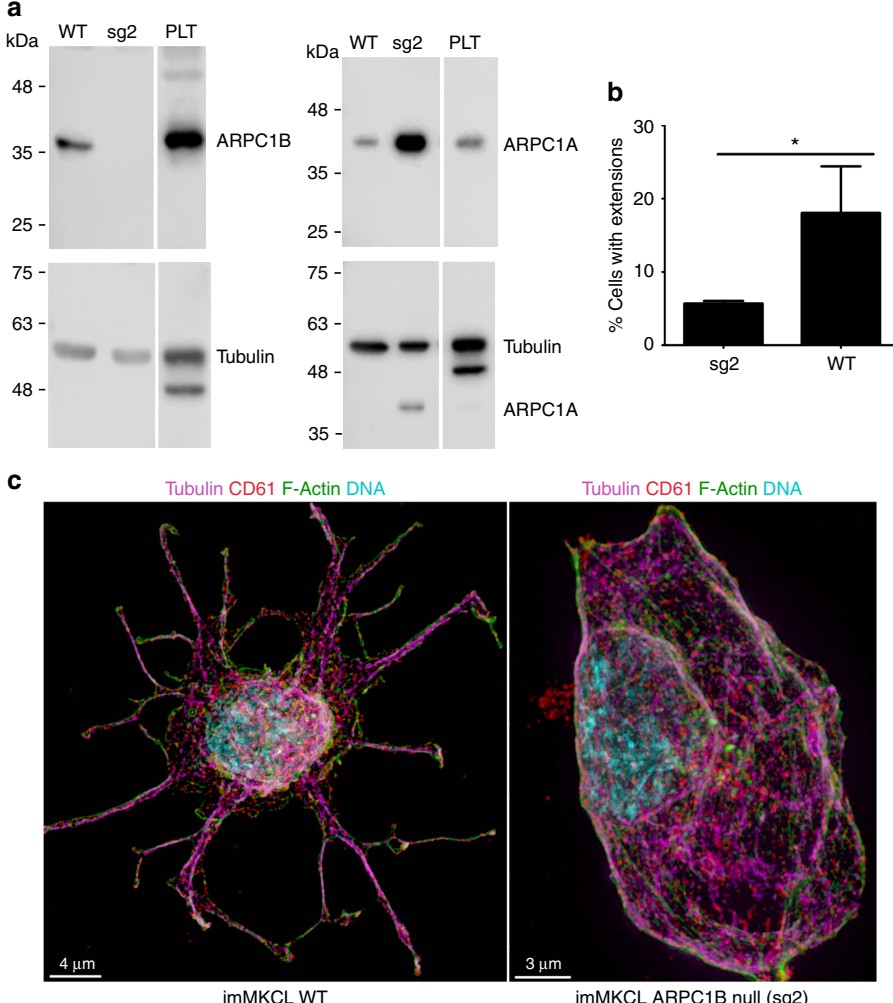

**Figure 9 | Decreased proplatelet formation in ARPC1B-null megakaryocytic cells.** (**a**) Cell lysate immunoblot analysis showing absent ARPC1B and increased ARPC1A expression in the ARPC1B-null imMKCL megakaryocytic cell line (sg2) relative to wild-type (WT) imMKCL cells and normal platelets (PLT; tubulin loading control for ARPC1B and ARPC1A indicated). (**b**) Wild-type and ARPC1B-null imMKCL cells were stimulated to differentiate in culture, and CD61-positive cells were scored for the presence of proplatelet-like extensions after 5 days; ARPC1B-null (sg2) cells showed a significantly decreased frequency of proplatelet formation compared to WT (*$P < 0.05$, two-tailed $t$-test; three experiments with 100 cells per group scored; bars show s.d.). (**c**) Comparisons of mature cultured representative wild-type imMKCL (left) and ARPC1B-null (right) cells; cells were stained and imaged by laser fluorescence structured illumination microscopy. Only cells expressing ARPC1B were observed to develop multiple structures resembling megakaryocyte proplatelets, containing tubulin and long, thin branched actin fibres (3D renders of images; bars = 4 μm left, 3 μm right).

Gel (NativePAGE, Novex, Invitrogen), electrophoresed and transferred on to PVDF membrane according to the manufacturer's recommendations. For two-dimensional gels (2D gels), individual gel lanes were cut, incubated in SDS–PAGE sample buffer containing β-mercaptoethanol for 30 min, and run in a 10% SDS-PAGE gel along with a MW marker and lysate sample. The gel was then blotted on to nitrocellulose membrane. For analysis of individual proteins lysates were diluted in reducing sample buffer and separated on 4–20% SDS–PAGE gels before blotting on nitrocellulose. Both PVDF and nitrocellulose membranes were probed with antibodies to Arp2/3 complex components (dilutions specified under Antibodies) and imaged via chemiluminescence in a Li-Cor Odyssey FC system; images were exported to Adobe Photoshop using Image Studio Lite software.

**Preparation of resting and fibrinogen spread platelets.** For resting platelets, PRP (above) was mixed with an equal volume of PBS plus 8% paraformaldehyde (PFA) and incubated for 10 min at room temperature. Fixed platelets were recovered via centrifugation, washed in PBS and resuspended at approximately 200 cells per nl in PBS plus 1% bovine serum albumin fraction V (BSA, MP Biomedicals), and 50 μl aliquots were spotted onto glass coverslips, incubated for 90 min at 37 °C with 100% humidity and rinsed with PBS. For platelet spreading, glass coverslips were treated with human fibrinogen (Sigma-Aldrich) for 30 min at 37 °C, then rinsed with PBS and allowed to dry. Aliquots of PRP or washed platelets (above) resuspended in Tyrode's solution (pH 7.2) were gently layered onto fibrinogen-treated surfaces and platelets were allowed to spread for

30–90 min. Adhered cells were then fixed with either 4% PFA (for IF) or with 2% glutaraldehyde (for electron microscopy), rinsed in appropriate buffers and stored at 4 °C with high humidity before preparation for imaging.

**High resolution fluorescence microscopy of platelets.** Preparations of resting/spread platelets were stained by permeabilizing/blocking with hybridization solution (PBS plus 2% donkey serum plus 1% BSA) containing 0.2% Triton-X100 for 60 min, rinsing with PBS and then incubating cells overnight at 4 °C with the appropriate primary antibodies. After washing, adherent antibodies were stained with Alexa Fluor-tagged donkey secondary antibodies (Thermo Fisher Scientific; Alexa Fluor 568 anti-mouse A10037, Alexa Fluor 488 anti-rabbit A10042, Alexa Fluor 647 anti-mouse A31571, Alexa Fluor 546 anti-goat A11056; dilutions 1/1000). Alexa Fluor tagged phalloidin (Thermo Fisher Scientific; dilution 1/300) was used to stain F-actin, Alexa Fluor tagged tubulin (Cell Signalling; dilution 1/300) was used to stain tubulin and Alexa Fluor tagged wheat germ agglutinin (dilution 10 μg ml$^{-1}$) was used to stain platelet surface membrane, all according to the manufacturer's protocols. Samples were mounted with ProLong Diamond or Gold antifade mountant (Thermo Fisher Scientific).

Spinning disc laser fluorescence confocal microscopy imaging was done with a Quorum Technologies system consisting of an Olympus IX81 inverted microscope, Hamamatsu C9100-13 back-thinned EM-CCD camera (512 × 512 pixels), Yokogawa CSU X1 spinning disk confocal scan head with Spectral Aurora Borealis upgrade, 4 diode-pumped solid state laser lines (Spectral Applied Research, 405,

491, 561, 642 nm), emission filters specific for Alexa Fluor dyes: 405 (447 ± 60 nm), 488 (525 ± 50 nm), 568 (593 ± 40 nm) and 647 (676 ± 29), and an ASI motorized XY stage controlled with an Improvision Piezo Focus Drive. Images were acquired with 250 nm Z-stepping via an Olympus UPLSAPO × 100/1.40 NA oil objective and a × 1.5 internal magnification lens (Spectral Applied Research) for a final magnification of × 150. Laser intensity, camera and exposure settings were established with minimal/undetectable levels of autofluorescence, channel crosstalk and non-specific primary/secondary background fluorescence. Acquisition, image deconvolution, registry correction (maximum one z-pixel required in our system) and cell surface area analysis were done with Volocity 6 software (Perkin-Elmer).

Laser fluorescence confocal structured illumination microscopy (SIM) was done using a Zeiss ELYRA PS.1 microscopy system (Axio Observer Z1 core) and a × 63/1.4 NA oil-immersion objective with × 1.6 optovar. The system is equipped with an Andor iXon3 885 detector, 405, 488, 561 and 640 nm laser lines, Zeiss motorized XY stage and Z-piezo focus. Acquisition control and SIM image processing (including channel alignment) were done with Zeiss Zen 2012 software using optimized settings and current calibration data sets. Rendered volume images were created from laser fluorescence confocal SIM and spinning disc confocal microscopy data using Imaris 8 software. Images were exported to Adobe Photoshop for labelling and presentation.

**Colon and skin immunofluorescence microscopy.** Paraffin sections of colon and skin were deparaffinized for 5 min each with: xylene twice, 100% ethanol twice, 95% ethanol twice, 75% ethanol, ddH2O. Antigen retrieval was performed with a Decloaking Chamber (NxGen) using a buffer containing 1.27 mM EDTA, 3.75 mM boric acid, 0.61 mM sodium borate and 0.003% ProClin. The slides were then blocked for 1 h (23 °C) with 5% BSA and 15% donkey serum in PBS. ARPC1B antibody (Sigma-Aldrich, HPA004832; dilution 1/100 in blocking buffer) kept overnight at 4 °C in a humidified staining chamber. After washing the slides three times with PBS containing 0.05% Tween 20, they were stained with Alexa Fluor 594 donkey anti-rabbit Fab fragment (Jackson ImmunoResearch; dilution 1/250) for 1 h at (23 °C). Slides were blocked with donkey anti-rabbit Fab fragment (Jackson ImmunoResearch; dilution 1/100) for 1 h (23 °C), then incubated with polyclonal rabbit anti-ARPC1A (Sigma-Aldrich, HPA004334; dilution 1/25) overnight at 4 °C. Secondary Alexa Fluor 488 donkey anti-rabbit (Jackson ImmunoResearch; dilution 1/250) was added for 1 h at RT. To minimize autofluorescence the slides were treated with 3.3 mM Sudan Black in 70% ethanol for 10 min (23 °C). Slides were then stained with DAPI (dilution 1/5,000) and mounted with Dako fluorescent mounting medium (S3023).

**Electron microscopy.** TEM of resting platelets was done as previously described[68]. Briefly, PRP was fixed with 2.5% glutaraldehyde in PBS and fixed overnight. Subsequently, platelets were post-fixed with 2% osmium tetroxide in $H_2O$ for 1 h and dehydrated in a graded series of acetone before embedding in Epon-Araldite. Thin sections were cut and stained with uranyl acetate and lead citrate. Grids were examined with a JEOL JEM-1011 electron microscope at 80 kV. Images were captured with a side-mounted Advantage HR CCD camera (Advanced Microscopy Techniques). Platelet whole-mount imaging was done by placing 2–3 drops of PRP on to Formvar-coated nickel grids (Electron Microscopy Sciences) for 5 min; excess liquid was removed with a filter paper followed by a 5-min fixation with 2.5% glutaraldehyde in PBS pH 7.4. After rinsing with distilled water the grids were placed into a JEOL JEM-1011 electron microscope with a 300/20 mm condenser/objective aperture. Dense granules were quantified by counting the number of dark spots in whole mounted platelets per TEM[29]. Granules were scored in a minimum of 50 platelets at × 15,000–100,000 magnification, and the mean number of dense granules per platelet was calculated.

Scanning electron microscopy was performed with samples of spread platelets fixed with glutaraldehyde (above) that were dehydrated, sputter coated with gold to 20 nm thickness in a Leica EM ACE200 high vacuum sputter coater and dried in a Bal-Tec CPD030 critical point dryer (32 °C, 75 bar). Imaging was done with a Philips XL-30 ESEM environmental scanning electron microscope.

**Generation of ARPC1B knockout imMKCL cells.** ARPC1B knockout imMKCL cells were generated via CRISPR/Cas9 knockout[69,70]. Four different sgRNA sequences complementary to ARPC1B were generated and cloned into lentiCRISPRV2: sgRNA1 5′-CAGACCGCAACGCCTACGTG, sgRNA2 5′-TCA CAATACGGTTACTCTCG, sgRNA3 5′-GCGTCCACACGTAGGCGTTG and sgRNA7 5′-GTTCACCTATGACGCCGCCG. HEK293T cells (ATCC, CRL-3216) were transfected with lentiCRISPRV2 constructs pMD2.G, pRSV-Rev and pMDLG/pRRE and lentiviral particles were collected for transduction of imMKCL megakaryocytic cells[49]. Non-transduced cells were eliminated by selection with puromycin and IB was used to assess surviving colonies for expression of ARPC1B and ARPC1A. The cell line generated with sgRNA2 showed loss of ARPC1B expression and upregulation of ARPC1A expression as observed in ARPC1B-null patient platelets (Fig. 9). ARPC1B-null and wild-type imMKCL cells were cultured and stimulated as previously described[49], and detailed here as follows. Both imMKCL WT and ARPC1B-null cells were grown in haematopoietic differentiation medium Iscove's modified Dulbecco's medium (IMDM) supplemented with human recombinant thrombopoietin (TPO, R&D systems #288-TP) 50 ng ml$^{-1}$,

human recombinant SCF (R&D systems #255-SC) 50 ng ml$^{-1}$ and doxycycline (Clontech, 631311) 1 μg ml$^{-1}$, in 37 °C incubator with 5% $CO_2$. The haematopoietic differentiation medium IMDM contained the following: IMDM (Life Technologies, 12440053), 15% fetal bovine serum, 10 × insulin/transferrin/selenite (Life technologies, 41400-045), 50 μg ml$^{-1}$ ascorbic acid (Sigma, A4544), 450 μM alpha-monothioglycerol (MTG, Sigma, M6145)

**Generation of proplatelets from imMKCL cultures.** On day 0 cells were collected into a 15 ml conical tube, centrifuged at 400$g$ × 5 min at 22 °C. After discarding the supernatant the cells were resuspended in 10 ml PBS and centrifuged at 400$g$ × 5 min at 22 °C. This washing step was repeated one more time. The cells were then resuspended in fresh haematopoietic differentiation medium supplemented with 50 ng ml$^{-1}$ human TPO, 50 ng ml$^{-1}$ human SCF and 15 μM ADAM17 inhibitor (TAPI-1, Sigma SML0739). The cells were then cultured in an incubator at 37 °C, 5% $CO_2$. On day 2, the cells were collected into a 15 ml conical tube, centrifuged at 400$g$ × 5 min at 22 °C. After removing the supernatant the cells were resuspended into fresh haematopoietic differentiation medium IMDM supplemented with 50 ng ml$^{-1}$ human TPO, 50 ng ml$^{-1}$ human SCF and 15 μM ADAM17 inhibitor. On day 3 the cells were seeded onto matrigel-coated coverslips. On culture day 5 cells were fixed, stained and imaged using protocols described above for spread platelets. CD61-positive cells were scored for the presence or absence of proplatelet extensions (Fig. 9).

**Data availability.** The whole exome sequencing data that support the findings of this study are available from the corresponding authors W.H.A.K and A.M.M on request. The data are not publicly available because they contain information that could compromise research participant privacy/consent. All other data generated or analysed during this study are included in this published article (and its Supplementary Information files) and available from the corresponding authors on request.

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

## Acknowledgements

The authors thank the patients and families described here from Canada. We thank Dr Koji Eto of the Center for iPS Cell Research and Application, Kyoto University, for generously supplying imMKCL cells. We also thank Karoline Fiedler, Lily Lu and Evelyn Uttama for their help. A.E. is supported by a Crohn's and Colitis Canada (CCC), Canadian Association of Gastroenterology (CAG) and Canadian Institute of Health Research (CIHR) Fellowship. C.T. and R.W.L. are supported by the RESTRACOMP fellowship from the Research Institute of the Hospital for Sick Children, Toronto, Canada. C.H.C. is supported by an Ontario Graduate Scholarship. W.H.A.K. is supported by operating grants from the Canadian Institutes of Health Research (CIHR; MOP-81208 and MOP-119450). A.M.M., R.S.M.Y. and J.H.B. are funded by a CIHR Team grant (THC 135233) in partnership with The Arthritis Society of Canada and Crohn's and Colitis Canada. C.M.R. is funded from the Jeffery Modell Foundation, Immunodeficiency Canada distinguished Professorship and the Canadian Centre for Primary Immunodeficiency. J.H.B. is the Pitblado Chair in Cell Biology at the Hospital for Sick Children. A.M.M. is funded by CIHR (MOP119457) and the Leona M. and Harry B. Helmsley Charitable Trust to study VEOIBD. We thank the interNational Early Onset Paediatric IBD Cohort Study (NEOPICS) for making the human data available.

## Author contributions

F.G.P., M.D., C.H.C., A.E., N.W., Q.L., C.T., J.P., G.L., I.L.-C., R.W.L., L.L., R.W.L. and R.M. carried out investigations under the supervision of W.H.A.K. and A.M.M. E.C., R.S.M.Y., J.U., R.M.L. and C.M.R. provided clinical care and edited the manuscript. W.H.A.K., F.G.P., J.H.B. and A.M.M. wrote the manuscript with contributions from all authors.

**Additional information**

**Competing interests:** The authors declare no competing financial interests.

