## [Peer Review File · Nature Communications]

Reviewers' comments:

Reviewer #1 (expert in platelets biology)

Remarks to the Author:

In their manuscript entitled, „Loss of the Arp2/3 complex component ARPC1B causes platelet abnormalities and inflammatory disease" Kahr and colleagues identified mutations within ARPC1B as a novel cause of microthrombocytopenia in humans. Their findings are novel and potentially interesting, but the presented data is in the present form not completely convincing and lack sufficient mechanistic insights and thus an explanation for the observed thrombocytopenia.

Major:

- The authors need to provide evidence that additional homozygous variants do not account for the observed defects. While the biochemistry data convincingly demonstrates absence of ARPC1B protein this does not prove that the mutation causes the disease.
- How do the authors explain that only patient 1 (c.269_270dupCT) suffered from thrombocytopenia, whereas patient 2 and 3 did not?
- The authors strongly focus on the observed microthrombocytopenia but did not address the causes of additional clinical features such as vasculitis, inflammatory bowel disease and eczema. This clearly weakens the study.
- What is the role of the elevated IgA? Could the increased IgA bind to platelets and modulate/induce their clearance?
- The quality of the transmission electron microscopy and immunofluorescence images in figure 3 appears inappropriate to be conclusive. While it is difficult to perform electron microscopy with very young thrombocytopenic patients, the quality of the presented images questions the validity of the conclusion on granule shape and appearance, particularly the quantification of dense granules, which are not detectable on the presented TEM images.
 - It would be great to present images of the bone marrow aspirates based on which MK morphology was rated to be normal. This could help to exclude or demonstrate a premature platelet release
- The authors should try to provide mechanistic insights by performing knockdown studies on CD34+ derived or iPS cells in order to address the cause of the observed microthrombocytopenia.

Reviewer #2 (expert in platelets biology)

Remarks to the Author:

In this study, Walter Kahr and colleagues provide new insights into the loss of the Arp 2/3 complex in platelet production and function abnormalities as well as inflammatory disease. In a series of groundbreaking experiments (which include a well-defined clinical work up, fluorescence and electron microscopy, western blots as well as genetic characterization), the authors clearly show that the Arp 2/3 complex plays a very important role in platelet production and platelet activation. Key findings include identification of a child with homozygosity for a frameshift mutation in ARPC1B and demonstration that ARPC1A cannot compensate. This is a really great paper because it is not only the first to show a key role for Arp 2/3 in the mechanisms of platelet production and activation, but it also provides physiological (in vivo) relevance as well as important clinical significance.

In general, the experiments are well designed and carefully executed and the manuscript is well written. The work is innovative and methods are state-of-the-art, thought provoking, will stimulate further research, and the subject will be of considerable interest to the readers of Nature Communications. The work is also based on strong rationale and has high clinical significance given the relevance of platelets to human health and disease. The work is comprehensive and contains a nice blend of hematology, immunology, and cytoskeleton biology. Because it connects many different

fields, I think that it is an extremely good fit for Nature Communications. Experiments on platelet morphology are carefully controlled and the data clearly supports their conclusions. I would only suggest the authors include more (higher magnification) electron microscopy images of the platelets. I would also suggest that authors include some more discussion of how their cell biological observations relate to the disease phenotype. In summary, this work represents a novel advance (this is a big finding!) for the field of platelet biology and the cytoskeleton field and helps to narrow down the list of proteins that may be involved in the molecular mechanisms of platelet production and activation.

Reviewer #3 (expert in genetics and platelets biology)
Remarks to the Author:

This work investigates mutations affecting ARPC1B in three human patients. Patient one was of southeast Asian descent and born of a consanguineous marriage. He had a homozygous mutation resulting in a frameshift of the ARPC1B gene. Patients 2 and 3 were brothers and the product of a supposedly non-consanguineous union. They were homozygous for two mutations in the ARPC1B gene. Patients 1 and 2 had in common symptoms of vasculitis with other symptoms specific to each. Patient 3 was homozygous for the identical mutations as his brother, patient 2, but had no significant clinical manifestations. Whole exome sequencing identified mutations in the ARPC1B gene for all three patients. Examination of the platelets found microthrombocytes and functional defects.

This work is an example of the power of using whole exome sequencing on patients for the discovery of potentially causative mutations and is thus novel. In addition, mutations in ARPC1B have never been associated with human disease.

The approaches taken by the authors are valid and the data are sound and the conclusions are generally consistent with the data presented. However, I have some suggestions that may improve this manuscript.

- 1) The lack of an overt vasculitis phenotype in patient 3 is puzzling given that he has the same homozygous ARPC1B mutations as his brother. This phenotypic heterogeneity should be discussed more fully. For example, does patient 3 have a different complement of variants than patient 2, which could potentially suppress the vasculitis phenotype? Have the other exonic mutations identified in patient 2 been investigated in patient 3?
- 2) If patients 2 and 3 are the product of a non-consanguineous marriage, how is it possible that they inherit the two identical closely-linked (and supposedly rare or unique) mutations from each parent?
- 3) In figure 1f, it would be useful to label the exons of the ARPC1B gene and the mutations inherited by each patient should be labeled. Also, the entire figure appears to be cropped.
- 4) The variable absence of the internal contents of the platelets in patients with no ARPC1B activity is very intriguing, could the authors provide more discussion of this point?

NCOMMS-16-00851A-Z

Response to Reviewer comments.

We thank all reviewers for their constructive comments. We have addressed all of these comments in our revised Manuscript, Figures and Supplementary Information, which has substantially improved our study and presentation of results. Our responses to specific reviewer comments are as follows:

Reviewer #1

Remarks to the Author:

In their manuscript entitled "Loss of the Arp2/3 complex component ARPC1B causes platelet abnormalities and inflammatory disease" Kahr and colleagues identified mutations within ARPC1B as a novel cause of microthrombocytopenia in humans. Their findings are novel and potentially interesting, but the presented data is in the present form not completely convincing and lack sufficient mechanistic insights and thus an explanation for the observed thrombocytopenia.

Response:

We appreciate that the reviewer found our data to be novel and potentially interesting. We have now addresses all the insightful comments by the reviewer, and as a result our manuscript is now much improved.

Major – individual questions addressed:

The authors need to provide evidence that additional homozygous variants do not account for the observed defects. While the biochemistry data convincingly demonstrates absence of ARPC1B protein this does not prove that the mutation causes the disease.

Response:

We have added a new paragraph in **Results: Identification of ARPC1B deficient patients** that provides a more extensive description of the process we used to home in on *ARPC1B* as the most likely sole source of the disease-associated variants identified in our patients. Specifically in Patient 1 there were no homozygous mutations in any known genes associated with platelet disorders or immune deficiency that could explain the phenotype. There were also no overlapping compound heterozygous, X-linked, *de novo* or autosomal dominant mutations shared between Patients 1 and 2. The critical role of Arp2/3 function made *ARPC1B* a suitable candidate gene for the WAS-like disease phenotype. This procedure is described in detail in the revised **Supplementary Methods: Identification of ARPC1B mutations** and the data presented in **Supplementary Tables 3A & B**. Since *ARPC1B* mutations are the only known/likely disease-associated gene variants shared by our patients, we are confident that these mutations are causative for their shared disease phenotype. For convenience we have provided the detailed analysis below:

Patient 1: Whole exome sequencing of Patient 1 (and parents) resulted in the identification of 50,020 variants. Of these, 31,997 variants were considered high quality, passing thresholds for genotype quality and read depth. 2,853 were considered rare variants with a minor allelic frequency less than 1% (MAF < 0.01) using three databases: NHLBI ESP6500SI-V2, 1000 Genomes and ExAC v.0.3). 1,046 variants were predicted to be damaging as they were classified as missense mutations or caused loss of function. From this list, we identified 14 non-synonymous homozygous variants (Supplemental Table 3A) that were inherited in an autosomal recessive manner. The *ARPC1B*, mutation resulted in a homozygous two base pair duplication in *ARPC1B* (c.269_270dupCT; Fig. 1f) that results in a frame shift in exon 4 at position 91 and a premature stop codon with a predicted truncated 128 amino acid protein (Fig. 1g). This gene was considered as a likely candidate as it is contained within a highly conserved region near a WD40 domain and was the only variant found to cause loss of function by RefSeq version 105v2. The c.269_270dupCT variant was validated using Sanger sequencing.

Patient 2: Whole exome sequencing of Patient 2 and 3 (and parents) resulted in the identification of 51,866 variants. Of these, 35,114 variants were considered high quality, passing thresholds for genotype quality and read depth. 1,704 were considered rare variants with a minor allelic frequency less than 1% (MAF < 0.01) using three databases: NHLBI ESP6500SI-V2, 1000 Genomes and ExAC v.0.3). 1,046 variants were predicted to be damaging as they were classified as missense mutations or caused loss of function. From this list, we identified 9 non-synonymous homozygous variants (**Supplementary Table 3B**) that were inherited in an autosomal recessive manner. Two homozygous *ARPC1B* missense variants were found near the same region of the mutation in Patient 1 (c.314C>T and c.712G>A encoding p.Ala105Val and p.Ala238Thr) in *ARPC1B* (Fig. 1g). Analysis of the *ARPC1B* variants from all patients using databases ExAC, NHLBI, ESP and 1000 Genomes confirmed that mutation c.269_270dupCT and variant c.314C>T were novel, whereas the variant c.712G>A was rare and predicted to be benign (**Supplementary Table 2**). Both variants were validated using Sanger sequencing.

Patient 1's parents are consanguineous and he has very severe and complex disease. Therefore we focused our initial genetic analysis on Mendelian autosomal recessive mutations in a homozygous inheritance pattern that could explain his disease. As shown in supplemental Tables 3A-B, there were no known homozygous mutations in predicted pathogenic variants in any known primary immune deficiency genes (including WASP and WIP) or platelet disorder genes that could explain the complex phenotype observed in Patient 1 and very similar phenotype observed in Patient 2. Furthermore, there were no overlapping compound heterozygote mutations, X-linked mutations, *de novo*, or autosomal dominant mutations identified shared between Patient 1 and 2 (see below figure). We then focused on novel genes and examined known biological function, known diseases associated with gene, gene expression profiles, and available animal models of all the candidates outlined in **Supplementary Table 3A & B**. The only gene that fit the disease observed in Patient 1 was *ARPC1B*. The critical role of *ARPC1B* in direct WASP binding and Arp2/3 regulation made it a viable candidate in the disease of Patient 1. Importantly, in Patient 2 with a very similar spectrum of disease, WES also identified *ARPC1B* as the only viable candidate (see also comments to Reviewer 3). Furthermore the only gene mutation that was common between both Patient 1 and 2 was *ARPC1B* (see Figure below).

How do the authors explain that only patient 1 (c.269_270dupCT) suffered from thrombocytopenia, whereas patient 2 and 3 did not?

Response:

We have expanded our explanation of patient phenotypic heterogeneity in the **Discussion** section. Patient 1 has complete absence of *ARPC1B* expression, associated with severe symptomatology

including thrombocytopenia. We attribute the milder phenotype and normal platelet counts in Patients 2 and 3 to residual ARPC1B expression. In addition, we have added new data showing that megakaryocytic imMKCL cells completely lacking ARPC1B protein are deficient in producing proplatelets (see below; **Supplementary Figure 8**). This suggests that loss of ARPC1B in patient megakaryocytes leads to decreased platelet production.

The authors strongly focus on the observed microthrombocytopenia but did not address the causes of additional clinical features such as vasculitis, inflammatory bowel disease and eczema. This clearly weakens the study.

Response:

We have expanded the discussion to elaborate on these additional clinical features. ARPC1B is critical for the function of Arp2/3, which modulates the actin dynamics that play a central role in hematopoiesis, immune system development, immune cell recruitment, migration and intercellular signaling, as well as activation of innate and adaptive immune responses. Identifying the molecular defect is the first step in the course of disease characterization, and determining the precise molecular role of ARPC1B and Arp2/3 complex function in these cellular processes will require further studies. Now included in the **Discussion** are several recent reports that demonstrate how Arp2/3 function is central to cellular processes important for innate and adaptive immune responses. We chose to focus on the abnormal platelet spreading phenotype because this allowed us to directly observe abnormal actin dynamics in the absence/deficiency of ARPC1B.

What is the role of the elevated IgA? Could the increased IgA bind to platelets and modulate/induce their clearance?

Response:

Elevated IgA has been noted in WAS, chronic infections, chronic liver disease, rheumatoid arthritis, SLE, sarcoidosis and multiple myeloma, suggesting that it is a marker of immune dysregulation and/or inflammation. We agree that IgA could in theory bind to platelets to enhance their clearance as in immune thrombocytopenia (ITP), however as ITP remains a diagnosis of exclusion (Cines et al. *Blood* 2009;113:6511-21) this would not be possible to assess in Patient 1. We have included a statement about potential antibody-mediated platelet clearance in our revised Discussion. It should be noted that while all patients had elevated IgA, platelet counts remained normal in Patients 2 and 3, suggesting that elevated IgA is insufficient to induce platelet clearance. It has also been reported (Falet et al. *Blood* 2009;114:4729-37) that in mice increased platelet-associated IgA protects WASP-interacting protein-deficient platelets from clearance. Thus while it may be interesting to pursue in the future, we are currently not able to assess whether elevated IgA is inducing platelet clearance in Patient 1.

The quality of the transmission electron microscopy and immunofluorescence images in figure 3 appears inappropriate to be conclusive. While it is difficult to perform electron microscopy with very young thrombocytopenic patients, the quality of the presented images questions the validity of the conclusion on granule shape and appearance, particularly the quantification of dense granules, which are not detectable on the presented TEM images.

Response:

To supplement the images shown in Figure 3 we have prepared a new **Supplementary Figure 2** which presents 12 panels of TEM images acquired at different magnifications. These images allow several platelets from Patients 1 and 2 to be compared to platelets from unaffected relatives. We agree that the quantitation of platelet dense granules cannot be made with thin section TEMs, given that there are only 3-8 dense granules per platelet (Reference #75 in manuscript). Thus we used whole-mount TEM to visualize and count calcium-rich dense granules (Supplementary Figure 4b). To emphasize and better illustrate this method, we have included representative platelet whole-mount TEM images in revised Supplementary Figure 4a. In addition we have now included the results of

lumi-aggregometry which revealed markedly decreased platelet dense granule ATP release in Patient 1 and 2. (see **Results: Morphology of ARPC1B deficient platelets**)

It would be great to present images of the bone marrow aspirates based on which MK morphology was rated to be normal. This could help to exclude or demonstrate a premature platelet release.

Response:

We have included new images of several megakaryocytes present in bone marrow aspirate preparations from Patient 1 in the revised Supplementary Figure 7. The morphology appears normal, as was confirmed by an expert hematopathologist at our institution who independently evaluated these bone marrow aspirates. As we mention in the revised text, this does not exclude the possibility of premature platelet release as this would be very difficult to assess in stained bone marrow samples. Determining premature platelet release would require obtaining CD34+ patient cells for semi-quantitative *in vitro* analysis, for which we do not have Research Ethics Board approval. We have, however, made significant progress evaluating proplatelet formation in megakaryocytic cells lacking ARPC1B, as described in the response to the next question.

The authors should try to provide mechanistic insights by performing knockdown studies on CD34+ derived or iPS cells in order to address the cause of the observed microthrombocytopenia.

Response:

We agree that such experiments would provide mechanistic insights into the observed cause of thrombocytopenia. We have considerable experience with introducing genes into primary CD34+ cells in culture using lentiviral-mediated transduction techniques (Noetzli et al. *Nat Genet* 2015 May;47(5):535-8). However, performing knockdown shRNA in such cells is challenging because knockdown is often incomplete, resulting in residual protein expression. Our goal was to create a phenocopy of Patient 1 megakaryocytes, which have no ARPC1B expression, in order to study the potential consequences for platelet development. In previous studies we have observed that iPS cells and most megakaryocytic cell lines do not recapitulate normal megakaryocyte development. Luckily, we were able to obtain immortalized megakaryocyte progenitor imMKCL cells from Dr. Koji Eto, which were created to generate platelets *in vitro* (Nakamura et al. *Cell Stem Cell* 2014;14(4):535-48). Experiments with these cells are presented in a new section of the **Results: Proplatelet formation in ARPC1B knockout megakaryocytic cells**, and in the new **Supplementary Figure 8**. Our results show that complete loss of ARPC1B expression significantly depresses proplatelet formation in imMKCL cells. As we now state in the revised **Discussion**, these observations indicate a link between loss of ARPC1B expression/Arp2/3 function and decreased proplatelet formation in megakaryocytes, which would have direct consequences for platelet production and thus cause the thrombocytopenia observed in our patients. Since terminal platelet production occurs in the bloodstream (Thon and Italiano, *Blood* 2012;120(8):1552-1561) – where pro-platelets trapped in microcapillaries are subjected to shear forces to drive final platelet production – we cannot draw any conclusions about the microthrombocytes observed in the patients using *in vitro* cultured MKs.

Reviewer #2

Remarks to the Author:

In this study, Walter Kahr and colleagues provide new insights into the loss of the Arp 2/3 complex in platelet production and function abnormalities as well as inflammatory disease. In a series of groundbreaking experiments (which include a well-defined clinical work up, fluorescence and electron microscopy, western blots as well as genetic characterization), the authors clearly show that the Arp 2/3 complex plays a very important role in platelet production and platelet activation. Key findings include identification of a child with homozygosity for a frameshift mutation in ARPC1B and demonstration that ARPC1A cannot compensate. This is a really great paper because it is not only the

first to show a key role for Arp 2/3 in the mechanisms of platelet production and activation, but it also provides physiological (in vivo) relevance as well as important clinical significance.

In general, the experiments are well designed and carefully executed and the manuscript is well written. The work is innovative and methods are state-of-the-art, thought provoking, will stimulate further research, and the subject will be of considerable interest to the readers of Nature Communications. The work is also based on strong rationale and has high clinical significance given the relevance of platelets to human health and disease. The work is comprehensive and contains a nice blend of hematology, immunology, and cytoskeleton biology. Because it connects many different fields, I think that it is an extremely good fit for Nature Communications. Experiments on platelet morphology are carefully controlled and the data clearly supports their conclusions.

Response:

We are pleased with the overwhelmingly positive evaluation of our manuscript.

I would only suggest the authors include more (higher magnification) electron microscopy images of the platelets.

Response:

We have included a new **Supplementary Figure 2** showing several TEM images of different fields at magnifications up to 40000X, comparing platelets from Patients 1 and 2 with unaffected relatives. We have also included new scanning electron microscopy (SEM) images of ARPC1B null and ARPC1B deficient fibrinogen-spread platelets (from Patients 1 and 2) in a new **Figure 4**.

I would also suggest that authors include some more discussion of how their cell biological observations relate to the disease phenotype.

Response:

We have described some of the potential mechanisms that could affect the disease phenotype in more detail in the **Discussion**. See also our response to Reviewer 1 regarding elaborating on the clinical features such as vasculitis, inflammatory bowel disease and eczema. We have also added new data describing experiments with imMKCL cells (**Supplementary Figure 8**), where we observed significantly decreased proplatelet formation in ARPC1B knockout imMKCL cells, implying a potential mechanism for the observed thrombocytopenia.

Reviewer #2 concludes: In summary, this work represents a novel advance (this is a big finding!) for the field of platelet biology and the cytoskeleton field and helps to narrow down the list of proteins that may be involved in the molecular mechanisms of platelet production and activation.

Response:

We are pleased to note that this reviewer finds our work as being novel and a big finding.

Reviewer #3

Remarks to the Author:

This work investigates mutations affecting ARPC1B in three human patients. Patient one was of southeast Asian descent and born of a consanguineous marriage. He had a homozygous mutation resulting in a frameshift of the ARPC1B gene. Patients 2 and 3 were brothers and the product of a supposedly non-consanguineous union. They were homozygous for two mutations in the ARPC1B gene. Patients 1 and 2 had in common symptoms of vasculitis with other symptoms specific to each. Patient 3 was homozygous for the identical mutations as his brother, patient 2, but had no significant clinical manifestations. Whole exome sequencing identified mutations in the ARPC1B gene for all three patients. Examination of the platelets found microthrombocytes and functional defects. This work is an example of the power of using whole exome sequencing on patients for the discovery of

potentially causative mutations and is thus novel. In addition, mutations in *ARPC1B* have never been associated with human disease. The approaches taken by the authors are valid and the data are sound and the conclusions are generally consistent with the data presented. However, I have some suggestions that may improve this manuscript.

Response:

We appreciate that this reviewer finds that the approaches taken by the authors are valid and the data are sound and the conclusions are generally consistent with the data presented.

1) The lack of an overt vasculitis phenotype in patient 3 is puzzling given that he has the same homozygous *ARPC1B* mutations as his brother. This phenotypic heterogeneity should be discussed more fully. For example, does patient 3 have a different complement of variants than patient 2, which could potentially suppress the vasculitis phenotype? Have the other exonic mutations identified in patient 2 been investigated in patient 3?

Response:

The revised manuscript and Supplementary material provides further details and expanded discussion of patient phenotypic heterogeneity. All 3 patients showed microthrombocytes, abnormal platelet spreading, growth failure and evidence of autoimmunity (eosinophilia, elevated IgA and IgE, positive ANCA; **Supplementary Table 1**). With regards to vasculitis, Patients 1 and 2 had a severe form while the eczema-like skin manifestation in Patient 3 has not been deemed severe enough to biopsy, thus we have been unable to determine if it is a milder form of skin disease. The revised manuscript states immunohistochemistry results showing lack of *ARPC1B* expression in skin biopsies.

With regards to other disease features, Patient 1 had a null mutation that resulted in a more severe phenotype including profound growth failure, chronic infections and an episode of colitis. As expected, Patients 2 and 3 had hypomorphic mutations that presented as a less severe form of the disease. This is similar to what is observed with *WASP* mutations that result in severe Wiskott-Aldrich syndrome (MIM301000), and less severe X-linked thrombocytopenia (MIM313900) and X-linked severe congenital neutropenia (*SCNX*; MIM300299). Furthermore, the differences in phenotypes observed between families 1 and 2, and between Patients 2 and 3 from family 2, are not unexpected as the majority of primary immune deficiencies, including Wiskott-Aldrich syndrome, exhibit phenotypic heterogeneity.

Regarding the question of different complements of variants and other exonic mutations in Patients 2 and 3 we have the following response. As shown by the laboratory parameters in **Supplementary Table 1** and discussed above, Patients 2 and 3 had the same mutations in *ARPC1B* (**Supplementary Table 2**). Although Patient 2 and 3 share many of the same features of disease, the variability of disease is not unexpected with almost all Mendelian genetic diseases including Wiskott-Aldrich syndrome. However, the clinical variability in disease with common mutations such as Delta508 in Cystic Fibrosis may be partially explained by modifier genes. In this case, with only 2 patients it is very difficult to identify modifying genes. We have not yet identified any genes that may explain the phenotype differences although this may be studied further.

2) If patients 2 and 3 are the product of a non-consanguineous marriage, how is it possible that they inherit the two identical closely-linked (and supposedly rare or unique) mutations from each parent?

Response:

This is a question we asked as well. The parents from Family 2 (Patients 2 and 3) did not self-describe as consanguineous. The family does, however, come from an isolated religious community in Canada. This community is confined to a limited genetic pool where familial aggregations of other described autosomal recessive and dominant conditions have been observed. Thus it is highly likely that there is a founder effect in this community and/or unreported consanguinity in this family. The family has not

provided any further detail and according to our ethical review board we are not allowed to further investigate any aspect other than the identification of causal variants.

3) In figure 1f, it would be useful to label the exons of the ARPC1B gene and the mutations inherited by each patient should be labeled. Also, the entire figure appears to be cropped.

Response:

We have revised figure 1f accordingly.

4) The variable absence of the internal contents of the platelets in patients with no ARPC1B activity is very intriguing, could the authors provide more discussion of this point?

Response:

This observation is now explained in more detail in the Discussion. WASP null platelets also show variable contents (**Figure 3e**), which suggests that actin dynamics are important in platelet formation, and possibly also in the maintenance of shape and internal contents by circulating platelets (i.e. WASP null and ARPC1B null platelets may be more fragile).

REVIEWERS' COMMENTS:

Reviewer #1 (Remarks to the Author):

The authors have carefully revised the manuscript which significantly improved the quality of the work. The paper now deserves to be published in a high ranking journal such as Nature Communications. I have no further comments.

Reviewer #2 (Remarks to the Author):

This is a top-notch manuscript describing how the loss of the Arp2/3 complex, an actin nucleation/branching regulator results in platelet abnormalities as well as predisposition to inflammation diseases. The revision is well-written, VERY novel, clear, and the data is high-level. Given the importance of Arp in the field of cell biology and platelets in health and disease, the work has very high significance. The expanded discussion on clinical features, the new electron microscopy, and the new knockdown studies on iPS-derived megakaryocytes have really improved the manuscript and have thoroughly addressed the concerns. I really like that this manuscript provides mechanistic connections between multiple fields. While I thought the initial manuscript was in the top 10%, the new data clearly pushed this work into the top 2%- no question. I highly recommend publication in Nature Communications.

Reviewer #3 (Remarks to the Author):

The manuscript has been adequately strengthened through revision.

One minor point regarding the discussion of ARPC1B deficiency being surprisingly nonlethal. From the ExAC database, it appears as if ARPC1B is not as constrained for loss of function variants as other members of the Arp2/3 complex, including ARPC1A. This may be due to the differential distribution of ARPC1A and B, (S. Table 5).

NCOMMS-16-00851B**Response to Reviewer comments.**

We thank all reviewers for their constructive comments. We have addressed the single minor issue raised by reviewer #3 in our revised Manuscript. Our responses to specific reviewer comments are as follows:

Reviewer #1**Remarks to the Author:**

The authors have carefully revised the manuscript which significantly improved the quality of the work. The paper now deserves to be published in a high ranking journal such as Nature Communications. I have no further comments.

Response:

We appreciate these remarks.

Reviewer #2**Remarks to the Author:**

This is a top-notch manuscript describing how the loss of the Arp2/3 complex, an actin nucleation/branching regulator results in platelet abnormalities as well as predisposition to inflammation diseases. The revision is well-written, VERY novel, clear, and the data is high-level. Given the importance of Arp in the field of cell biology and platelets in health and disease, the work has very high significance. The expanded discussion on clinical features, the new electron microscopy, and the new knockdown studies on iPS-derived megakaryocytes have really improved the manuscript and have thoroughly addressed the concerns. I really like that this manuscript provides mechanistic connections between multiple fields. While I thought the initial manuscript was in the top 10%, the new data clearly pushed this work into the top 2%- no question. I highly recommend publication in Nature Communications.

Response:

We are honored by the overwhelmingly positive evaluation of our revised manuscript.

Reviewer #3**Remarks to the Author:**

The manuscript has been adequately strengthened through revision. One minor point regarding the discussion of ARPC1B deficiency being surprisingly nonlethal. From the ExAC database, it appears as if ARPC1B is not as constrained for loss of function variants as other members of the Arp2/3 complex, including ARPC1A. This may be due to the differential distribution of ARPC1A and B, (S. Table 5).

Response:

We appreciate that the reviewer finds our manuscript adequately strengthened through revision. We have addressed the minor point raised by rephrasing the questionable statement "ARPC1B deficiency being surprisingly nonlethal" The sentence has been changed to reflect the conclusions of a paper published in *J Cell Biol* (Molli et al. 2010 Jul 12;190(1):101-14) where we now state: "It has been reported that ARPC1B is both an activator and substrate of Aurora A kinase which is critical in the maintenance of mitotic integrity in mammalian cells".